# MRMR: A Realistic and Expert-Level Multidisciplinary Benchmark for Reasoning-Intensive Multimodal Retrieval

**Siyue Zhang**[*,N,S]   **Yuan Gao**[*,J]   **Xiao Zhou**[*,J]   **Yilun Zhao**[Y]   **Tingyu Song**[A]
**Arman Cohan**[Y]   **Anh Tuan Luu**[N,V]   **Chen Zhao**[S,C]

[N]Nanyang Technological University   [Y]Yale University   [S]NYU Shanghai
[J]Shanghai Jiao Tong University   [A]University of the Chinese Academy of Sciences
[C]Center for Data Science, New York University   [V]VinUniversity

## Abstract

We introduce MRMR, the first expert-level multidisciplinary multimodal retrieval benchmark requiring intensive reasoning. MRMR contains 1,435 queries spanning 23 domains, with positive documents carefully verified by human experts. Compared to prior benchmarks, MRMR introduces three key advancements. First, it challenges retrieval systems across diverse areas of expertise, enabling fine-grained model comparison across domains. Second, queries are reasoning-intensive, with images requiring deeper interpretation such as diagnosing microscopic slides. We further introduce Contradiction Retrieval, a novel task requiring models to identify conflicting concepts. Finally, queries and documents are constructed as image–text interleaved sequences. Unlike earlier benchmarks restricted to single images or unimodal documents, MRMR offers a realistic setting with multi-image queries and mixed-modality corpus documents. We conduct an extensive evaluation of 4 categories of multimodal retrieval systems and 14 frontier models on MRMR. The text embedding model Qwen3-Embedding with LLM-generated image captions achieves the highest performance, highlighting substantial room for improving multimodal retrieval models. Although latest multimodal models such as Ops-MM-Embedding perform competitively on expert-domain queries, they fall short on reasoning-intensive tasks. We believe that MRMR paves the way for advancing multimodal retrieval in more realistic and challenging scenarios. [1]

## 1 Introduction

LLM-based agents, such as DeepResearch (OpenAI, 2024; Qiao et al., 2025), have been widely applied in domains including science, engineering, medicine, and finance (Zhao et al., 2025; Tang et al., 2024; Barry et al., 2025; Phan et al., 2025). These systems move beyond the intrinsic knowledge of LLMs by actively retrieving and integrating external information, making a strong and robust retrieval component essential (Chen et al., 2025). In practice, many expert-domain applications rely on multimodal information, underscoring the need for retrieval methods that can handle queries and documents spanning both visual and textual modalities, or even interleaved image–text content (Zhang et al., 2021; Liu et al., 2021; 2023). For instance, given a medical image, the agent system should retrieve similar cases or guidelines to support clinical decisions.

While existing multimodal retrieval benchmarks have made progress, they are insufficient to capture the complexity of agentic scenarios. We identify three key limitations: (1) **Multidisciplinary expert domains**: most multimodal benchmarks are built on Wikipedia text and images, focusing on general-domain knowledge (Hu et al., 2023; Chen et al., 2023; Zhang et al., 2025b). However, state-of-the-art LLMs already demonstrate strong capabilities in handling such knowledge (Team et al., 2025), making it essential to develop benchmarks for high-stakes expert domains such as medicine, science,

---

[1]Our data is available at `https://huggingface.co/datasets/MRMRbenchmark`.

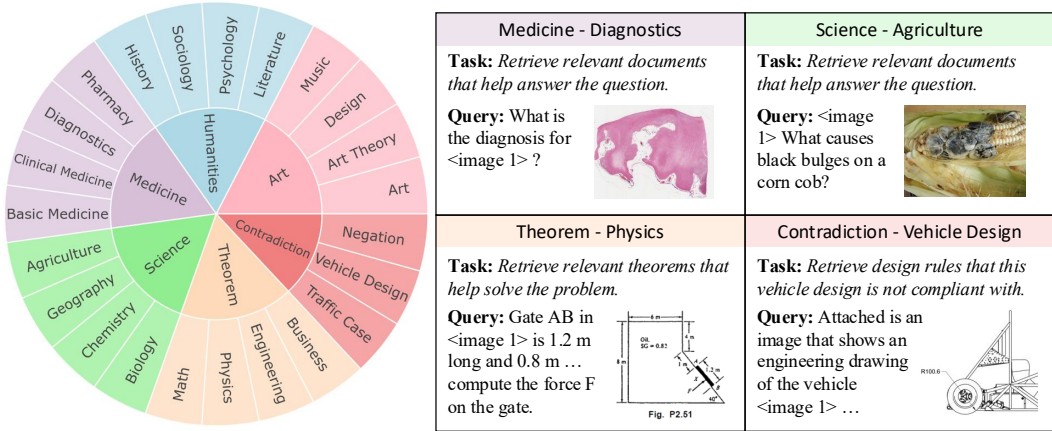

Figure 1: Overview of the MRMR benchmark. MRMR includes 1,435 expert-annotated examples, covering 23 domains across 6 disciplines. It is specifically designed to assess multimodal retrieval models in expert-level, reasoning-intensive tasks. Notably, we originally introduce the *Contradiction Retrieval* task in the multimodal setting, which requires retrieving documents that conflict with the user query and features deeper logical reasoning.

and engineering. (2) **Reasoning intensity**: existing benchmarks primarily target semantic matching and information-seeking tasks, whereas real-world queries often involve expert-domain images and require deeper understanding and logical reasoning over them. (3) **Image-text interleaving**: prior benchmarks mostly support single-image queries with supplementary text, yet real-world queries and documents typically consist of interleaved text and multiple images (Zhang et al., 2025b).

To address these gaps, we introduce **MRMR**, a comprehensive benchmark measuring retrieval models in expert-level **M**ultidisciplinary and **R**easoning-intensive **M**ultimodal **R**etrieval. Figure 1 presents an overview of our benchmark. MRMR consists of 1,435 expert-annotated examples, categorized into three types of retrieval tasks: (1) *Knowledge* for retrieving web pages related to queries involving multiple expert-domain images; (2) *Theorem* for retrieving theorems involved in solving multimodal math problems; and (3) *Contradiction* for retrieving contradictory statements or rules given a case description. Specifically, we derive complex multidisciplinary queries from established Visual Question Answering (VQA) benchmarks (Yue et al., 2024; 2025) and assign expert annotators to collect positive documents from Internet. To build a sizable corpus, we additionally include negative documents from knowledge-intensive collections (Wang et al., 2024a; Su et al., 2025). To further elevate the reasoning challenge, we originally introduce *Contradiction Retrieval*, which requires models not only to detect semantic relevance but also to perform logical reasoning to identify conflicting concepts. To foster a deeper integration of visual and textual content, we represent both queries and documents in an interleaved multimodal format.

We conduct an extensive evaluation on MRMR across four main categories of multimodal retrieval approaches and 14 representative models. The results reveal that current multimodal retrieval systems consistently underperform text-only retrievers with image captioning on knowledge- and reasoning-intensive multimodal queries. The highest score of 54.1 is achieved by the text embedding model Qwen3-Embedding (Zhang et al., 2025d) combined with LLM-based image captioning. The best-performing multimodal model, Ops-MM-Embedding (OpenSearch-AI, 2025), trails by 6.0 points, mainly due to its limited reasoning capabilities rather than domain expertise. Its performance drops from 67.4 on *Knowledge* tasks to 37.4 and 36.6 on *Theorem* and *Contradiction* tasks, even though the corpora for these two tasks are much smaller than that of *Knowledge*. More importantly, the multidisciplinary setup in MRMR reveals substantial performance differences across models and domains. For instance, Ops-MM-Embedding surpasses the second-best model, MM-Embed (Lin et al., 2025), in the Art discipline, whereas their performances are comparable in the Medicine discipline. We hope our benchmark and findings will help progress in multimodal retrieval.

Table 1: Comparison of multimodal retrieval benchmarks and MRMR. In the "Modality" column, "T → I" indicates retrieving image documents using a text query. The "#Domain" column reports the number of domains; "Open" denotes datasets built from Wikidata with general domains. The "Expert", "Reason", and "Interleaved" columns indicate whether expert knowledge is required, whether intensive reasoning is involved, and whether data are in the interleaved image-text format.

| Benchmarks | Modality | Retrieval Type | #Domain | #Query | Expert? | Reason? | Interleaved? |
|---|---|---|---|---|---|---|---|
| NIGHTS | I → I | Visual Similarity | Open | 20K | ✗ | ✗ | ✗ |
| SciMMIR | T ↔ I | Image Caption | 11 | 530K | ✓ | ✗ | ✗ |
| EDIS | T → IT | Image Caption | Open | 3,241 | ✗ | ✗ | ✗ |
| Wiki-SS | T → I | Document QA | Open | 3,610 | ✗ | ✗ | ✗ |
| WebQA | T → IT | Document QA | Open | 2,511 | ✗ | ✗ | ✗ |
| ViDoRe | T → IT | Document QA | 10 | 3,810 | ✓ | ✗ | ✗ |
| MMDocIR | T → IT | Document QA | 10 | 1,658 | ✓ | ✗ | ✗ |
| FashionIQ | IT → I | Composed Image | 1 | 12,238 | ✗ | ✗ | ✗ |
| CIRR | IT → I | Composed Image | Open | 4,148 | ✗ | ✗ | ✗ |
| CIRCO | IT → I | Composed Image | Open | 1,020 | ✗ | ✗ | ✗ |
| InfoSeek | IT → IT | VQA | Open | 1.35M | ✗ | ✗ | ✗ |
| OVEN | IT → IT | VQA | Open | 18,341 | ✗ | ✗ | ✗ |
| wikiHow-TIIR | IT → IT | VQA | Open | 7,654 | ✗ | ✗ | ✓ |
| MRMR | IT → IT | VQA | 23 | 1,435 | ✓ | ✓ | ✓ |

## 2 RELATED WORK

**Benchmarking multimodal retrieval.** As illustrated in Table 1, existing multimodal retrieval datasets mainly focus on semantic matching or information-seeking tasks. Early semantic matching benchmarks are built from paired image–text data, where the text is semantically aligned with the image (Liu et al., 2023; Wu et al., 2024; Xiao et al., 2025; Jiang et al., 2025c), and the task is to retrieve the corresponding modality. Composed Image Retrieval (CIR) emerges as a challenging task that allows users to search for target images using a multimodal query, comprising a reference image and a modification text specifying the user's desired changes to the reference image (Zhang et al., 2021; Liu et al., 2021; Baldrati et al., 2023; Zhang et al., 2024). Information-seeking benchmarks either retrieve supporting evidence for visual questions (Hu et al., 2023; Chen et al., 2023) or retrieve multimodal documents for textual queries (Ma et al., 2024; Macé et al., 2025; Dong et al., 2025). As all prior studies focus on single-image inputs, TIIR (Zhang et al., 2025b) proposes a more realistic setup in which the query and document consist of interleaved text–image sequences supporting multiple images. However, it is limited to searching general-domain wikiHow tutorials. To further advance multimodal retrieval, we construct MRMR, the first benchmark comprising complex multidisciplinary queries that require in-depth reasoning in the interleaved text–image format.

**Multimodal retrieval models and multimodal retrieval augmented generation.** State-of-the-art multimodal retrieval models commonly rely on large pre-trained encoders such as CLIP (Radford et al., 2021) and BLIP (Li et al., 2023), which map images and texts into a shared embedding space. Their outputs are often combined using fusion strategies (*e.g.,* score fusion) to integrate information across modalities (Wei et al., 2024). More recent works adapt multimodal large language models (MLLMs) for universal multimodal embeddings by fine-tuning them on diverse retrieval tasks (Jiang et al., 2025b; Zhang et al., 2025c; Jiang et al., 2025c; Lin et al., 2025). In these approaches, multimodal queries are processed through the MLLM, and the hidden states from the final transformer layer, typically the last token representation, are used as the dense embedding for retrieval. In this work, we benchmark a diverse set of multimodal retrieval approaches, including text retrievers with image captioning, text and image two-stream models with vector fusion, and multimodal retrievers. Additionally, thanks to advances in both retriever and generative models, multimodal retrieval-augmented generation (MM-RAG) has emerged as a key application (Hu et al., 2025; Jiang et al., 2025a; Wu et al., 2025b; Zhan et al., 2025; Wasserman et al., 2025). While various MM-RAG benchmarks have been introduced, most focus on evaluating response generation and lack evidence-level relevance annotations, making it impractical to assess retrieval performance and its contribution within MM-RAG (Chen et al., 2025).

Table 2: Data statistics of MRMR. For each dataset, we show the number of queries ($Q$) and documents ($D$), the average number of positive documents ($D_+$) per example, the average number of text tokens of queries and documents (measured by the GPT-2 tokenizer (Radford et al., 2019), not including task instruction text), the average number of images in queries and documents, and sources of queries and documents. *Knowledge* datasets share a common retrieval corpus, while *Theorem* datasets share another. Examples for each dataset can be found in Appendix G.

| Dataset | Total Number | | | Avg. #Text | | Avg. #Images | | Source | | Ex. |
|---|---|---|---|---|---|---|---|---|---|---|
| | $Q$ | $D$ | $D_+$ | $Q$ | $D$ | $Q$ | $D$ | $Q$ | $D$ | |
| *Knowledge* | | | | | | | | | | |
| Art | 157 | 26,223 | 1.8 | 15.4 | 421.6 | 1.1 | 0.72 | MMMU-Pro knowledge question | PIN-14M, Web pages | Fig. 13 |
| Medicine | 167 | 26,223 | 2.2 | 32.0 | 421.6 | 1.1 | 0.72 | | | Fig. 14 |
| Science | 137 | 26,223 | 1.8 | 32.1 | 421.6 | 1.2 | 0.72 | | | Fig. 15 |
| Humanities | 94 | 26,223 | 1.9 | 54.5 | 421.6 | 1.2 | 0.72 | | | Fig. 16 |
| *Theorem* | | | | | | | | | | |
| Math | 60 | 14,257 | 2.1 | 64.6 | 364.3 | 1.1 | 0.001 | MMMU-Pro calculation question | BRIGHT, Web pages | Fig.17 |
| Physics | 104 | 14,257 | 2.1 | 56.2 | 364.3 | 1.0 | 0.001 | | | Fig.18 |
| Engineering | 190 | 14,257 | 2.0 | 53.5 | 364.3 | 1.0 | 0.001 | | | Fig.19 |
| Business | 158 | 14,257 | 3.2 | 64.2 | 364.3 | 1.0 | 0.001 | | | Fig.20 |
| *Contradiction* | | | | | | | | | | |
| Negation | 200 | 4 | 1.0 | 0.0 | 12.8 | 1.0 | 0.00 | COCO | Synthetic | Fig.21 |
| Vehicle Design | 88 | 700 | 1.0 | 152.5 | 107.5 | 1.0 | 0.04 | DesignQA | Design Rules | Fig.22 |
| Traffic Case | 80 | 796 | 1.8 | 19.5 | 123.3 | 1.0 | 0.58 | Synthetic | Driving Handbook | Fig.23 |

**Reasoning-intensive retrieval.** Beyond keyword- and semantic-based information retrieval, BRIGHT (Su et al., 2025) has introduced the first benchmark in the text domain that requires intensive reasoning to identify relevant documents. For example, given a new math or physics problem, the retrieval system is expected to provide previously solved problems using the same theorems or relevant theorem statements. To tackle this challenge, recent methods train the text retrievers using synthetic datasets containing complex queries and hard negatives (Weller et al., 2025; Das et al., 2025; Zhang et al., 2025a; Long et al., 2025; Shao et al., 2025; FlagEmbedding, 2025). Our work extends reasoning-intensive retrieval into the multimodal domain. MRMR is constructed by sourcing expert-level queries from the multimodal understanding and reasoning benchmark MMMU-Pro (Yue et al., 2025), collecting image-text interleaved documents from web pages, and obtaining relevance annotations from human experts.

## 3 MRMR BENCHMARK

### 3.1 TASK FORMULATION

We define the task of multimodal retrieval as follows. Let $Q = \{q_1, \ldots, q_n\}$ be the set of queries and $D = \{d_1, \ldots, d_m\}$ the document corpus. Each query $q$ and document $d$ is represented as a sequence of segments $(x_1, \ldots, x_k)$, where each segment $x$ can be either text or an image. For a query $q$, a document can be either a positive document $d_+$ (relevant) or a negative document $d_-$ (non-relevant). In reasoning-intensive retrieval, a document $d$ is considered relevant if it provides principles or theorems that support the reasoning chain required to answer query $q$ (Su et al., 2025). Unlike prior studies (Xiao et al., 2025; Dong et al., 2025), we do not constrain the corpus to uniform data types, reflecting more realistic retrieval scenarios. To evaluate diverse reasoning capabilities, we design three types of retrieval tasks in MRMR:

- *Knowledge*. It emphasizes reasoning over broad expert domain knowledge. For a multimodal query, a document is relevant if expert annotators confirm that it contributes to reasoning about the query by providing critical concepts or theoretical foundations.

- *Theorem*. It targets the theorem-based reasoning over calculation problems. For a multimodal calculation query, a document is relevant if it conveys the same underlying theorem or formula needed to solve the problem.

- ***Contradiction.*** It requires logical reasoning to detect conflicting or inconsistent concepts. For a multimodal case description query, a document is relevant if it provides the rule or requirement that the query violates.

## 3.2 KNOWLEDGE: RETRIEVING WEB PAGES THAT HELP ANSWER QUESTIONS

MMMU (Yue et al., 2024) is one of the most widely used benchmark for evaluating multi-discipline multimodal understanding in MLLMs. Its robust version, MMMU-Pro (Yue et al., 2025), excludes questions solvable by text-only models, expands the candidate options, and provides verified correct answers. We repurpose the knowledge- and reasoning-intensive questions in MMMU-Pro as queries $Q$ and construct a corpus $D$ of image–text interleaved documents. The positive documents $D_+$ are scraped from relevant websites referenced by the GPT-Search[2] model (OpenAI, 2024) and verified by human experts; while negative documents $D_-$ are augmented by sampling from the multimodal collection PIN-14M (Wang et al., 2024a) (see Figure 2).

**Selecting questions.** We prompt GPT-5[3] to categorize MMMU-Pro questions into two groups, *i.e.,* knowledge-based and calculation questions. We adopt calculation questions for the *Theorem* subset in Section 3.3. For knowledge questions, we then instruct GPT-5 to filter out questions that require only superficial reasoning over text and images, without reliance on external domain expertise. For the remaining questions, we generate detailed descriptions for each associated image using GPT-5, which we include as part of the input context for subsequent steps.

**Constructing positive and hard negative documents.** Unlike keyword- or semantic-based multimodal retrieval benchmarks, collecting positive documents for our queries is more time-consuming because it requires identifying and validating multimodal sources that support the query's answer. To address this, we design a semi-automated pipeline with human expert annotators. Specifically, for each query, given the GPT-5-generated image descriptions and ground-truth answer, we prompt GPT-Search to reason over the question and generate an explanation for the correct answer with reference web links pointing to diverse materials such as Wikipedia, books, academic papers, and blogs. To preserve the completeness of multimodal content, we capture these webpages as PDFs, apply MonkeyOCR (Li et al., 2025) to extract interleaved text and images, and split the content into chunks while preserving image references. Resulting documents are then screened by GPT-5 and validated by human experts about whether they support the correct answer. Documents with GPT-human agreement on relevance are retained as positives, those agreed irrelevant as hard negatives, while ambiguous cases (30–60% across domains) are discarded. In cases where GPT-Search fails to retrieve relevant documents (38.2% of questions), expert annotators are instructed to search the web and create one supporting document, optionally including image links within the text. Due to the complexity of the questions, the number of positive documents per query is typically fewer than four. We annotate data anonymously through the Turkle platform (HLT-COE@JHU, 2025), with detailed guidelines provided in Appendix B.

**Constructing additional negative documents.** After the previous step, we obtain 993 cleaned and annotated documents for 555 queries. To construct a sizable retrieval corpus comparable to (Xiao et al., 2025; Su et al., 2025), we supplement these with negative documents sampled from the large-scale multimodal collection PIN-14M (Wang et al., 2024a), which contains knowledge-intensive resources such as medical articles from PubMed Central (PMC)[4] and web content from OBELICS (Laurençon et al., 2023). Given the wide topic coverage and large number of documents in PIN-14M, we assume a low probability of false negatives for our sampled documents. We validate this assumption through manual error analysis in Section 5.1. In total, we curate a corpus of 26,223 documents, including text only, image only, and text-image interleaved.[5]

---

[2]GPT-Search refers to the version `gpt-4o-search-preview-2025-03-11` throughout this work.

[3]GPT-5 refers to the version `gpt-5-2025-08-07` throughout this work.

[4]https://www.ncbi.nlm.nih.gov/pmc/

[5]The corpus could be further expanded by sampling additional expert-domain documents, which naturally increases retrieval difficulty and the probability of false negatives. We leave it as future work.

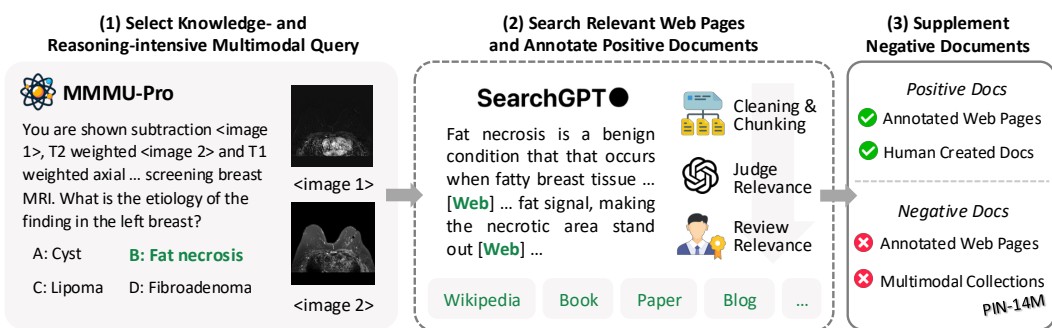

Figure 2: An overview of the data construction workflow for MRMR (*Knowledge*). We select and convert knowledge- and reasoning-intensive questions from MMMU-Pro (Yue et al., 2025) into retrieval queries. Web pages such as Wikipedia, blogs, and papers referenced by the GPT-Search model during reasoning are processed into documents through screen capturing, OCR (Li et al., 2025), and chunking. The relevance of resulting documents is first evaluated by GPT and then verified by expert annotators. Lastly, we source negative documents from the knowledge-intensive multimodal collection PIN-14M (Wang et al., 2024a) to construct a sizable corpus.

### 3.3 THEOREM: RETRIEVING RELEVANT THEOREMS THAT SOLVE PROBLEMS

As introduced by Su et al. (2025), retrieving relevant theorem statements can assist users in solving new math or physics problems. We extend this formulation to the multimodal domain by leveraging challenging calculation problems from MMMU-Pro. In this setting, the query $q$ is a image-centric calculation problem, and the corpus $D$ consists of theorem descriptions across domains such as mathematics, physics, engineering, and business. A document $d$ is regarded as positive if it describes a theorem applicable to solving the query problem.

**Selecting questions.** From the calculation questions in MMMU-Pro, we first use GPT-5 to exclude questions that explicitly state the required theorem in the text. The remaining questions are then organized into four major domains: Math, Physics, Engineering, and Business. The Engineering domain further includes areas such as Mechanical Engineering, Computer Science, and Electronics, while the Business domain covers Finance, Economics, Marketing, and related fields. Then, we prompt GPT-5 to reason through each multimodal question, produce a final answer, and summarize the key theorems used in the solution. We exclude questions for which GPT-5 produces incorrect answers, with a final set of 512 questions.

**Constructing positive and negative documents.** We adopt the theorem statements from BRIGHT (Su et al., 2025) as the primary retrieval corpus (∼13.8k documents), reflecting the realistic setting where most theorems are expressed in text. For each question, the summarized key theorems are used as queries to retrieve the top-10 candidate statements from the corpus with the Qwen3-Embedding model (Zhang et al., 2025d). Among these candidates, GPT-5 identifies the most relevant theorem statements, which are retained as positive documents, while the rest serve as negatives.

**Constructing additional positive documents.** Not all theorems have relevant counterparts in BRIGHT. To address this, we scrape additional theorem statements, optionally accompanied by illustrative images, from webpages such as Wikipedia, following the OCR pipeline described in Section 3.2. GPT-5 then rewrites these theorems to match the format of BRIGHT statements. Finally, we deduplicate the scraped documents to ensure a consistent and complete retrieval corpus. Consequently, 63.6% of the positive documents are sourced from webpages, with the remainder drawn from the BRIGHT corpus. More details are presented in Appendix C.

### 3.4 CONTRADICTION: RETRIEVING CONTRADICTORY RULES AND REQUIREMENTS

Most existing datasets emphasize retrieving positively supporting evidence for a query (Xiao et al., 2025; Chen et al., 2023; Dong et al., 2025). However, retrieving contradictory information could be of great importance especially in expert domains. For example, a user may provide a case description

and seek evidence of violation of laws, policies, or guidelines, as shown in Figure 23. In this setting, the query $q$ is a case description (*e.g.,* traffic or vehicle design cases), while the corpus $D$ comprises mandated rules (*e.g.,* driving theory handbooks or design requirements). A document $d$ is considered positive if it contains the statement or rule contradicting the query case. Unlike traditional retrieval tasks, this new formulation requires not only semantic matching between query and document but also deep logical reasoning to identify conflicting concepts.

**Negation.** To study contradiction retrieval, we first design a synthetic task inspired by the negation benchmark NegBench (Alhamoud et al., 2025). Given an image from COCO (Lin et al., 2014) with ground truth object annotations, we synthesize four candidate text descriptions: three accurately reflecting the objects and one containing a contradiction, either by asserting the existence of a non-existent object or the absence of an existent one. The models are required to pinpoint the text description conflicting with the given image in a multi-choice setup. For example, in Figure 21, the query image shows a keyboard on the table, while the positive document explicitly states that none is present, revealing a contradiction. More details are provided in Appendix D.1.

**Vehicle Design.** To evaluate contradiction retrieval in engineering documents, we construct a vehicle design task by leveraging the Formula SAE Rulebook and design cases from the DesignQA dataset (Doris et al., 2025). In industrial product design, designers must review hundreds of pages of requirement documents to ensure their designs comply with specifications. To assist designers, retrieval systems are expected to identify the specific sections that a design case fails to satisfy. For example, in Figure 22, the vehicle's wheelbase in the design is shorter than the required minimum, indicating a contradiction. During data preparation, we introduce variations to the design cases and chunk the lengthy design document, as detailed in Appendix D.2.

**Traffic Case.** Retrieval systems have been applied to legal documents to assist legal professionals in preparing arguments and citations (Feng et al., 2024). To evaluate this capability in multi-modality, we construct a traffic case task to assess whether retrievers can identify which driving rules are violated in traffic cases. We build the corpus by chunking official driving handbooks (Singapore Police Force, 2017) into sections. Meanwhile, we build the query set by selecting dozens of driving rules, each linked to several annotated violation cases. We augment these violation cases by replacing key textual elements with AI-generated images using Qwen-Image (Wu et al., 2025a). For example, as shown in Figure 23, a car is driving only 3 meters behind the vehicle ahead — significantly less than the required safe distance. Further details are provided in Appendix D.3.

## 4 EXPERIMENTS

### 4.1 EXPERIMENTAL SETUP

We evaluate 4 types of multimodal retrieval setups with 14 frontier models, as follows: (1) **Text models with image caption (T2T)**: We assess text retrievers, namely BGE-M3 (Chen et al., 2024), NV-Embed-V2 (Lee et al., 2025), and Qwen3-Embedding-8B (Zhang et al., 2025d), by pairing with MLLM-generated image captions (see Appendix E.1 for details). (2) **Text and image two-stream models with vector fusion (IT2IT)**: We evaluate CLIP-style two-stream models, including EVA-CLIP (Sun et al., 2023), SigLIP (Zhai et al., 2023), OpenCLIP (Cherti et al., 2023), and JinaCLIP (Koukounas et al., 2024), by a simple vector-fusion strategy. Given an input sequence, we concatenate all text chunks for one text embedding $t$, while all images are concatenated vertically for another image embedding $i$. Following MTEB (Xiao et al., 2025), the final score is computed using the fused embedding $e = t + i$. (3) **Multimodal models with merged image (IT2IT)**: We evaluate multimodal retrievers including VISTA (Zhou et al., 2024), E5-V (Jiang et al., 2025b), MM-Embed (Lin et al., 2025), VLM2Vec (Jiang et al., 2025c), Ops-MM-Embedding (OpenSearch-AI, 2025) and GME-Qwen2-VL (Zhang et al., 2025c). Since these models support only single-image input, multiple images are concatenated in the same way as for two-stream models. (4) **Multimodal models with document as image (T2I)**: We also include the document retrieval paradigm that receives text-only query and encode entire multimodal documents as screenshot images, such as ColPali (Faysse et al., 2025). Because these models are trained for text queries, query images are replaced with LLM-generated captions, similar to the text retriever setup. Besides, we note that a native image–text interleaved model, TIIR (Zhang et al., 2025b), has been introduced and is expected to

Table 3: The performance of retrieval models on MRMR. We report nDCG@10 for all subtasks except Negation, for which we use Hit@1: Art, Medicine (Med.), Science (Sci.), Humanities (Hum.), Math, Physics (Phy.), Engineering (Eng.), Business (Bus.), Negation (Neg.), Design, and Traffic. Avg. denotes the average score across 11 subtasks. The best score on each subtask is highlighted in **bold**, and the second best is underlined.

| | Knowledge | | | | Theorem | | | | Contradiction | | | Avg. |
|---|---|---|---|---|---|---|---|---|---|---|---|---|
| Model | Art | Med. | Sci. | Hum. | Math | Phy. | Eng. | Bus. | Neg. | Design | Traffic | |
| *Text Models with Image Caption* (T2T) | | | | | | | | | | | | |
| BGE-M3 | 48.6 | 30.0 | 42.4 | 45.6 | 16.5 | 19.5 | 21.6 | 39.3 | 16.0 | 25.9 | 17.4 | 29.3 |
| NV-Embed-v2 | 70.7 | 46.8 | 65.7 | 66.6 | 26.4 | 35.2 | 32.9 | 52.2 | 12.5 | 42.1 | 42.2 | 44.8 |
| Qwen3-Embedding | 71.9 | **53.2** | **72.5** | **74.4** | **37.7** | **50.2** | **42.9** | **58.3** | 12.0 | **67.8** | **54.2** | **54.1** |
| *Text and Image Two-Stream Models with Vector Fusion* (IT2IT) | | | | | | | | | | | | |
| EVA-CLIP | 10.2 | 13.5 | 26.1 | 12.9 | 6.2 | 12.2 | 10.7 | 17.4 | 8.5 | 4.4 | 5.4 | 11.6 |
| SigLIP | 26.7 | 14.7 | 26.7 | 12.3 | 7.4 | 6.5 | 5.9 | 12.5 | 13.5 | 4.9 | 9.6 | 12.8 |
| OpenCLIP | 56.0 | 17.9 | 33.2 | 22.0 | 7.5 | 6.6 | 7.3 | 14.0 | 13.0 | 8.1 | 12.4 | 18.0 |
| JinaCLIP | 21.4 | 16.8 | 27.1 | 10.7 | 10.9 | 7.5 | 9.1 | 13.7 | 10.5 | 16.5 | 9.7 | 14.0 |
| *Multimodal Models with Merged Image* (IT2IT) | | | | | | | | | | | | |
| VISTA | 21.3 | 27.8 | 32.6 | 17.0 | 18.8 | 17.1 | 17.3 | 28.6 | 20.0 | 20.2 | 9.4 | 20.9 |
| E5-V | 25.1 | 11.7 | 16.6 | 10.8 | 2.1 | 3.4 | 2.5 | 5.2 | 11.5 | 3.7 | 2.1 | 8.6 |
| MM-Embed | 65.6 | 53.0 | 63.5 | 62.8 | 23.6 | 30.8 | 27.4 | 44.9 | 7.0 | 23.8 | 34.9 | 39.8 |
| VLM2Vec | 53.5 | 22.4 | 36.7 | 24.0 | 2.1 | 2.8 | 2.8 | 2.9 | 11.5 | 5.6 | 18.3 | 18.1 |
| GME-Qwen2-VL | 54.3 | 40.1 | 46.8 | 45.6 | 28.8 | 36.0 | 30.2 | 45.1 | 15.0 | 26.3 | 29.6 | 36.2 |
| Ops-MM-Embedding | **79.3** | 52.5 | 70.0 | 67.8 | 27.7 | 39.5 | 30.1 | 52.3 | 8.0 | 55.9 | 45.8 | 48.1 |
| *Multimodal Models with Document as Image* (T2I) | | | | | | | | | | | | |
| GME-Qwen2-VL | 54.0 | 40.7 | 59.0 | 50.1 | 21.2 | 22.1 | 27.0 | 45.3 | 14.5 | 56.1 | 40.1 | 39.1 |
| Ops-MM-Embedding | 67.7 | 48.8 | 67.7 | 63.9 | 25.0 | 34.0 | 29.2 | 49.0 | 10.5 | 59.8 | 46.3 | 45.6 |
| ColPali | 36.1 | 29.9 | 42.7 | 29.2 | 7.3 | 17.5 | 13.5 | 34.6 | **28.5** | 19.4 | 18.2 | 25.2 |

best fit the interleaved format of MRMR; however, it is not publicly available. We provide details of each model in Appendix E.1. Following prior work (Xiao et al., 2025; Su et al., 2025), we use nDCG@10 as the main evaluation metric except Negation. Since each query in Negation has exactly one gold document among four candidates, we adopt Hit@1 as the main metric for this task.

## 4.2 MAIN RESULTS

**Multimodal retrieval systems lag behind text retrieval-based approaches on knowledge- and reasoning-intensive images.** As shown in Table 3, the text retriever Qwen3-Embedding combined with LLM-based image captioning achieves the highest performance (54.1 nDCG@10). Although captions may omit certain visual details, they provide rich contextual descriptions and additional knowledge that substantially benefit retrieval. In contrast, multimodal systems struggle with the expert-level query images in MRMR, which often require deep reasoning, such as diagnosing microscopic tissue sections (Figure 1). CLIP-style two-stream models are particularly limited, as their training emphasizes alignment of superficial text–image semantics and model sizes are relatively small. The most recent MLLM-based embedding models, such as Ops-MM-Embedding, show promising results under both interleaved text–image and document-as-image paradigms, indicating the effectiveness of unified training on diverse retrieval tasks.

**Multimodal retrieval systems perform particularly poorly on reasoning-intensive tasks.** While Ops-MM-Embedding achieves a solid 67.4 nDCG@10 on *Knowledge* subtasks, its performance drops sharply to 37.4 and 36.6 on *Theorem* and *Contradiction*, respectively. Models such as E5-V and VLM2Vec perform even worse, essentially failing on these tasks. This gap highlights the difficulty of extracting abstract concepts from practical problems, for example linking an image-based physics question to the relevant theorem in Figure 1. Notably, Hit@1 scores for most models on the synthetic *Contradiction* task Negation remain below 25%—equivalent to random guessing given four candidates per query. As illustrated in the Negation example Figure 21, humans can readily detect conflicting concepts embedded within supporting evidence, yet retrieval models struggle even for strong text embedding models. Although the candidate corpora for the Design and Traffic

subtasks are much smaller than those of standard knowledge bases (Su et al., 2025; Dong et al., 2025), models still struggle to identify the underlying contradictions. Nevertheless, surface-level semantic matching remains useful in these settings, as it allows models to locate relevant documents without fully resolving the conflicting concepts (*e.g.,* a query concerning driving speed matched with a document specifying the speed limit). These findings suggest that current retrieval models possess strong capabilities in semantic matching and information seeking, but remain fundamentally limited in their reasoning ability.

**Substantial differences in performance are evident across models and domains.** Across all four multimodal retrieval settings, we observe a wide performance difference between models. For instance, among multimodal models with merged image, the weakest model, E5-V, achieves only 8.6 nDCG@10, whereas Ops-MM-Embedding reaches 48.1 nDCG@10, revealing substantial methodological differences. As MRMR is the first multidisciplinary multimodal retrieval benchmark, it enables fine-grained domain-level evaluation. For example, as shown in the breakdown performance Table 7, MM-Embed performs competitively with Ops-MM-Embedding in medical domains such as Clinical Medicine and Diagnostics, yet lags behind in art-related tasks. We also observe pronounced variation in retrieval difficulty across domains. In the Art subtasks, systems can often succeed by matching query images to visually identical or similar artworks, which narrows the search space. However, in medical imaging, such overlap is rare, and models are required to identify underlying pathological and radiological features rather than relying on superficial visual similarity.

## 5 ANALYSIS

### 5.1 QUALITATIVE ANALYSIS

To understand model limitations, we conduct 30 case studies by manually reviewing their top-10 documents retrieved by Ops-MM-Embedding. There are two major failure patterns that we have observed. (1) **Visual bias over contextual relevance**: in the Agriculture case as shown in Figure 11, the model ranks a negative document higher because it contains a nematode SEM image resembling the earthworm image in the query, even though the positive document provides a detailed discussion of the key topic Fauna. Similar errors occur in Medicine, where visually similar eye images from different diseases mislead the model. (2) **Failure of higher-level deduction**: in the Traffic case as shown in Figure 12, the model assigns a higher score to a negative document than to a positive one because both depict cars, tunnels, and lane markings. However, it fails to infer that the car is crossing the line, which contradicts the positive document's instruction to "Stay in lane". Although multimodal retrievers exhibit these shortcomings and lag behind text-only retrievers with image captions, we believe they remain essential because many real-world queries inherently span across modalities. Fundamentally, textual descriptions alone cannot fully capture the nuanced information in images, especially when MLLMs lack the required visual knowledge.

### 5.2 TEST-TIME SCALING IN RETRIEVAL

Query expansion is a widely used technique, recently framed as test-time scaling in retrieval (Shao et al., 2025). Prior work (Su et al., 2025) demonstrates that incorporating explicit reasoning substantially improves performance on reasoning-intensive text retrieval tasks. Motivated by this, we have conducted comparative experiments to evaluate the effectiveness for multimodal retrieval. Specifically, we prompt MLLMs, including Qwen2-VL-2B-Instruct (Wang et al., 2024b) and Qwen2.5-VL-72B-Instruct (Bai et al., 2025), to generate reasoning traces, including question summarization and chain-of-thought reasoning, following (Su et al., 2025). As shown in Table 9, replacing the original queries with MLLM-generated reasoning traces leads to substantial performance improvements: +5.1 for Qwen2-VL-2B and +14.8 for Qwen2.5-VL-72B. The improvements are particularly pronounced on *Knowledge* tasks, whereas *Theorem* tasks benefit to a lesser extent. Meanwhile, we observe that, without constraining output length, the larger model Qwen2.5-VL-72B produces on average 20% and 66% more tokens than Qwen2-VL-2B in *Knowledge* and *Theorem* respectively, trading higher inference cost for larger performance gains (see more details in Appendix F.3).

## 6 CONCLUSION

We introduce MRMR, a realistic, multidisciplinary, reasoning-intensive multimodal retrieval benchmark. We leverage knowledge- and reasoning-intensive questions from MMMU-Pro and build a sizable multimodal corpus with positive documents verified by human experts. In addition, we introduce Contradiction Retrieval for evaluating models' logical reasoning capabilities to identify conflicts. Comprehensive evaluation shows that multimodal retrieval systems lag behind their text-retrieval counterparts, indicating substantial room for improvement. Although state-of-the-art multimodal models excel in *Knowledge* domains, they drop nearly 30 points on reasoning-intensive tasks. We hope MRMR facilitates identifying model limitations and advancing multimodal retrieval.

## CODE OF ETHICS AND ETHICS STATEMENT

All data used in constructing MRMR are sourced from publicly available materials and are employed solely for academic research, not commercial use. We have carefully ensured that the dataset contains no private information or harmful content, such as discriminatory, violent, or unethical material. Our goal is to support socially beneficial research. Following the practice of MMMU (Yue et al., 2024), our annotators and validators are instructed to avoid using materials from websites that prohibit copying or redistribution when reviewing MRMR documents. Consequently, most documents are derived from sources that are free of copyright restrictions, such as Wikipedia pages, government reports (e.g., National Institutes of Health and Singapore Police Force), and PubMed Central (PMC). The datasets we build upon also carry permissive public licenses, including MMMU (Apache-2.0), PIN-14M (CC-BY-4.0), COCO (CC-BY-4.0), and BRIGHT (CC-BY-4.0). For test-time scaling, we primarily focus on text expansion rather than image resizing and process as the text expansion has shown more significant impacts.

## REPRODUCIBILITY

Our datasets and annotation process are introduced in Section 3, and the experimental settings are described in Section 4. Specific implementation details are provided in Appendix E.1. To facilitate the reproduction of our experiments, the data are available at `https://huggingface.co/datasets/MRMRbenchmark`, and the evaluation code is provided at `https://github.com/rebeccaz4/MRMR`.

## ACKNOWLEDGMENTS

This research is supported by the RIE2025 Industry Alignment Fund – Industry Collaboration Projects (IAF-ICP) (Award I2301E0026), administered by A*STAR, as well as supported by Alibaba Group and NTU Singapore through Alibaba-NTU Global e-Sustainability CorpLab (ANGEL). Siyue Zhang and Chen Zhao were supported by NYU Shanghai Center for Data Science. This work was supported in part through the NYU IT High Performance Computing resources, services, and staff expertise.

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

# Appendix Contents

# A  THE USE OF LARGE LANGUAGE MODELS

In this work, large language models (LLMs) are employed solely as tools for data generation, as described in the main paper. Importantly, no parts of the manuscript are generated by LLMs. Hence, there are no concerns of plagiarism or scientific misconduct related to text generation.

# B  DATASET CONSTRUCTION: KNOWLEDGE

## B.1  ANNOTATOR BIOGRAPHY

The detailed biographies of the annotators involved in MRMR construction are presented in **Table 4**. All annotators are from universities ranked in the Top 500 of the 2025 QS Global Rankings[3] and are fluent in English. Annotators assess document–query relevance by judging whether a document facilitates answering the query. To ensure quality, independent validators conduct an additional round of verification.

Table 4: Biographies of 24 annotators involved in MRMR construction (Author biographies are hidden to protect identity confidentiality).

| ID | Year | Major | Assigned Subject(s) | Author? | Validator? |
|---|---|---|---|---|---|
| 1 | 3rd year Undergraduate | Biological Engineering | Biology | ✗ | ✗ |
| 2 | 1st year Master | Biological Engineering | Biology | ✗ | ✓ |
| 3 | 1st year Master | Biomedical Engineering | Biology, Pharmacy | ✗ | ✗ |
| 4 | 2nd year Master | Biomedical Engineering | Biology, Pharmacy | ✗ | ✗ |
| 5 | 1st year Master | Biomedical Engineering | Biology, Pharmacy | ✗ | ✗ |
| 6 | 1st year PhD | Chemistry | Chemistry | ✗ | ✗ |
| 7 | 2nd year Master | Chemistry | Chemistry | ✗ | ✓ |
| 8 | 3rd year PhD | Medicine | Basic Medicine | ✗ | ✗ |
| 9 | 3rd year Undergraduate | Clinical Medicine | Clinical Medicine, Diagnostics | ✗ | ✗ |
| 10 | 3rd year Undergraduate | Medicine | Basic Medicine | ✗ | ✓ |
| 11 | 2nd year Master | Clinical Medicine | Clinical Medicine, Diagnostics | ✗ | ✗ |
| 12 | 2nd year Master | Clinical Medicine | Clinical Medicine, Diagnostics | ✗ | ✓ |
| 13 | 3rd year Undergraduate | Pharmacology | Pharmacy | ✗ | ✓ |
| 14 | 4th year Undergraduate | Pharmacology | Pharmacy | ✗ | ✗ |
| 15 | 1st year Master | Music | Music | ✗ | ✗ |
| 16 | 1st year Master | Clinical Medicine | Clinical Medicine | ✗ | ✗ |
| 17 | 1st year PhD | Sociology | Sociology, Psychology | ✗ | ✗ |
| 18 | 1st year Master | Bioinformatics | Biology | ✗ | ✗ |
| 19 | 2nd year PhD | Agricultural and Biosystems Engineering | Agriculture | ✗ | ✗ |
| 20 | 4th year Undergraduate | Literature | History, Literature | ✗ | ✗ |
| 21 | 3rd year Undergraduate | Geography and Environmental Studies | Geography | ✗ | ✗ |
| 22 | 4th year PhD | Computer Science | - | ✓ | ✓ |
| 23 | 4th year Undergraduate | Computer Science | - | ✓ | ✓ |
| 24 | 3rd year Undergraduate | Electronic Engineering | - | ✓ | ✓ |

## B.2  ANNOTATION GUIDELINE AND INTERFACE

To facilitate data annotation, we develop the following interface based on Turkle (HLT-COE@JHU, 2025), an open-source clone of Amazon's Mechanical Turk. The annotation guideline and interface is detailed in Figure 3, Figure 4, and Figure 5.

## B.3  DATA ANNOTATION PAYMENT

The annotation and validation process for MRMR spanned three months. Each annotator was assigned approximately **50 questions** aligned with their academic major. After annotation, validators independently assessed the quality of the labels. We provided a *base rate* of **7 USD per hour**, with a quality adjustment of about 10%. On average, annotating a single question required **10 minutes**, while validation took **4 minutes**. This compensation scheme ensured that annotators received wages competitive with the average teaching assistant salary at their universities. To maintain a manageable workload and reduce pressure, we recommended a maximum of **10 questions per day**.

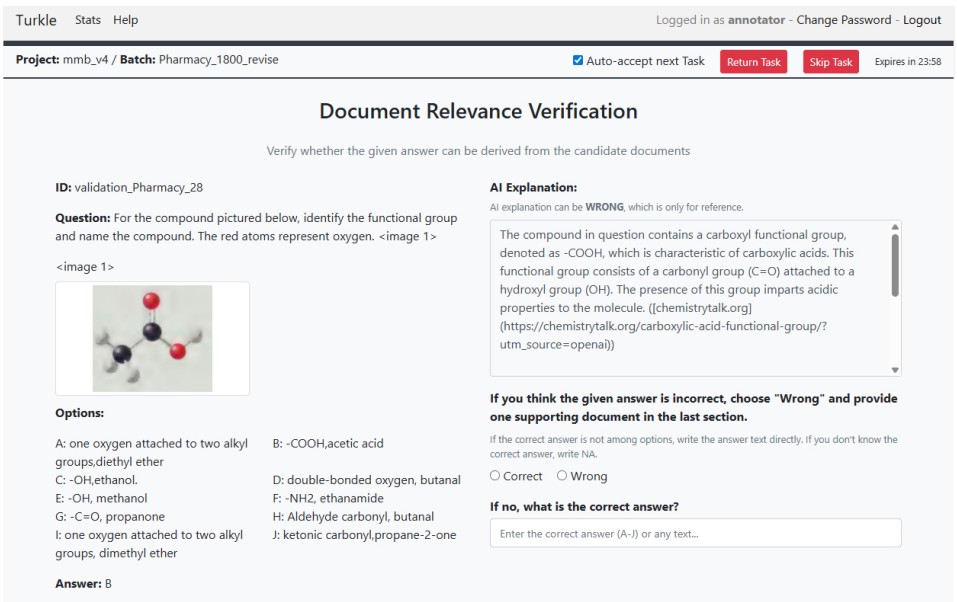

Figure 3: **Annotation Interface - Step 1: Question Understanding.** Annotators are first shown the question, associated images, candidate options, the correct answer, and an AI-generated explanation. The explanation is provided to aid understanding, though annotators are informed it may be incorrect. In this step, they judge whether the given answer is correct based on their own knowledge.

### B.4 DATASET CONSTRUCTION PROMPTS

The dataset construction prompts are presented in Figure 6, Figure 7, Figure 8, and Figure 9.

## C DATASET CONSTRUCTION: THEOREM

### C.1 THEOREM DATABASE CONSTRUCTION

The BRIGHT theorem corpus was embedded using Qwen3-Embedding (Zhang et al., 2025d) and indexed in ChromaDB, which supports efficient semantic search via HNSW (Chroma, 2025). Each entry retains a unique `theorem_id` and the original `text`, enabling fast, semantics-aware retrieval with full traceability to the source.

### C.2 WIKIPEDIA CONTENT PROCESSING PIPELINE

We retrieved Wikipedia content by querying the MediaWiki Search API (MediaWiki, 2024) using theorem names as search keys. For supplementary sources in PDF format, we employed Monkey-OCR (Li et al., 2025) to convert scanned documents into Markdown. The resulting text was then processed through a structured extraction prompt (Figure 10) using GPT-5 to perform final cleaning, normalization, and precise theorem statement extraction.

### C.3 DOCUMENT DEDUPLICATION

All theorems extracted from Wikipedia were deduplicated prior to inclusion in the corpus. Deduplication was performed in two stages: first by theorem name, and then by semantic content using TF-IDF–based cosine similarity (Salton & Buckley, 1988). Specifically, we employed `TfidfVectorizer` to compute `TF-IDF` vectors for all theorem statements (Pedregosa et al., 2011), followed by pairwise cosine similarity. Entries with near-identical content (cosine similarity $\geq 0.85$) were collapsed into a single representative instance.

     Back to Appendix Table of Contents

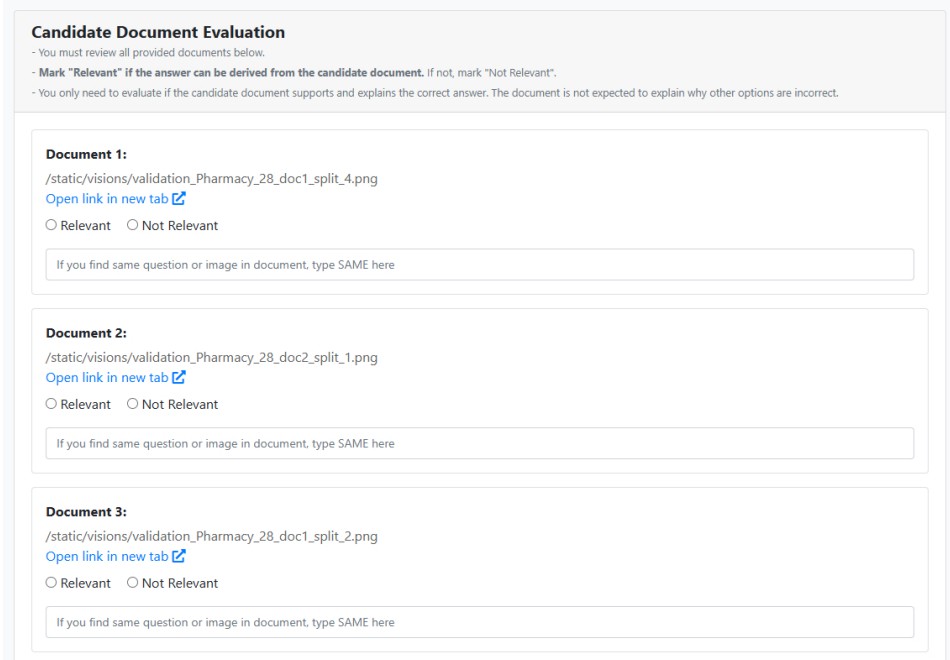

Figure 4: **Annotation Interface — Step 2: Candidate Document Evaluation.** After understanding the question, annotators are instructed to review candidate documents individually and judge whether each can facilitate correctly answering the question. Documents are shown in image format, with up to eight candidates presented. Document relevance definition has been explained to annotators before the annotation process.

# D  DATASET CONSTRUCTION: CONTRADICTION

## D.1  NEGATION

First, we randomly select 200 samples from the COCO (Lin et al., 2014) dataset, each containing at least three positive objectives. For each entry, we construct a description using the template, "The image includes $a$, $b$, $c$, but no $d$." In the positive description, we randomly select three positive objectives to replace $a$, $b$, and $c$, and select one negative objective to replace d. For the negative description, we generate two variations: one where all four objectives ($a$, $b$, $c$, $d$) are selected from the positive objectives, and another where one of $a$, $b$, or $c$ is replaced by $a$ randomly selected negative objective. The image from each sample is used as the query, and the three positive descriptions and one negative description are used as the corpus. Finally, we manually review the 200 queries and corresponding gold documents to ensure that the contradictory descriptions are identifiable by humans, and revise any ambiguous queries for clarity. No LLM prompting is involved in constructing the Negation task.

## D.2  VEHICLE DESIGN

On one hand, to construct the queries, we use design cases from the DesignQA dataset (Doris et al., 2025) and augment them through appropriate modifications, such as altering numerical values and introducing variations in image elements. On the other hand, to construct the corpus, we apply MonkeyOCR (Li et al., 2025) to extract and segment the Formula SAE Rulebook into 700 files, organized by rule ID. Finally, we review all the queries to ensure they represent incorrect designs.

## D.3  TRAFFIC CASE

First, we select a set of traffic rules and, based on these rules, create traffic violation cases by crafting relevant stories. These stories are then used as prompts to generate 12 images for each story using

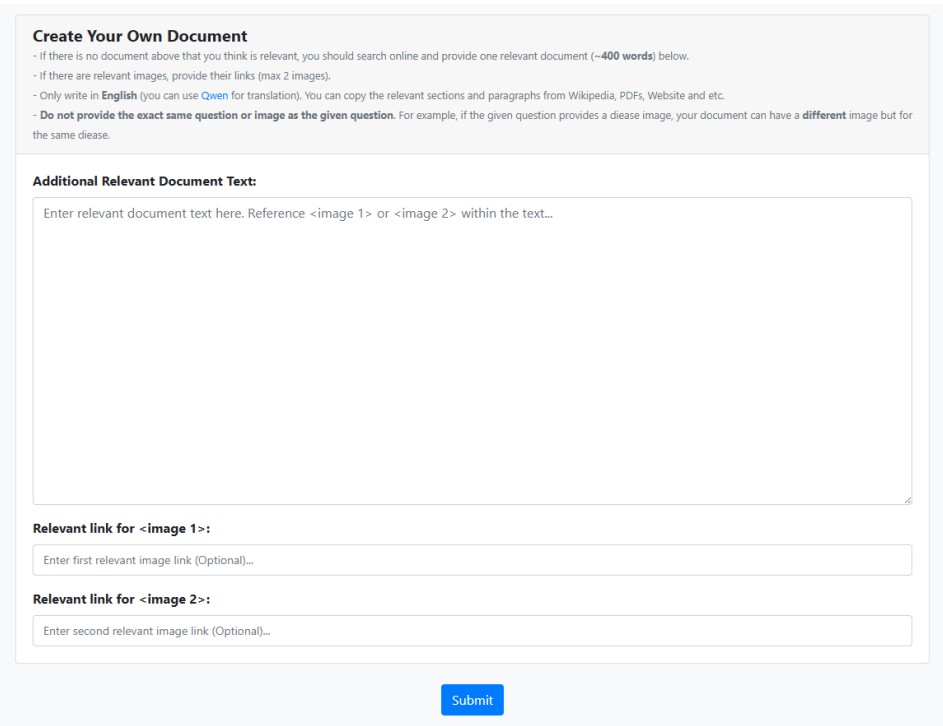

Figure 5: **Annotation Interface — Step 3: Create Relevant Document.** If none of the candidate documents are deemed relevant, annotators are required to search for a suitable web page and provide the gold evidence content. They are encouraged to include images from the source, and the final document is written in an interleaved image–text format.

Your task is to determine whether a question with images requires expert knowledge, such as about a historical event, scientific concept, economic theory, or medical disease. The last line of your response should be of the following format: "Result: YES_OR_NO" (without quotes). If the answer can be obtained easily by reading the question text and image content alone without the need of expert knowledge, say NO. Think step by step before answering. Here are some examples:

`{example_1}`

`{example_2}`

Now please determine whether this new question requires expert knowledge:

`{question_and_answer}`

Figure 6: The prompt for determining whether the question is knowledge-based.

GPT-5. Afterwards, we manually review all the generated images and use Doubao (Gong et al., 2025) to refine and enhance them for better clarity and relevance. Additionally, we leverage Doubao to generate specific objectives from the queries in order to construct image–text interleaved queries. For the corpus, we use MonkeyOCR to split Basic Theory of Driving and Final Theory of Driving (Singapore Police Force, 2017), two official driving handbooks in Singapore, into separate files, which are then organized and used as the corpus. Finally, we conduct a manual review of all the queries, ensuring that any additional corpus IDs caused by excessive image details are properly incorporated into the queries.

Is the multimodal question testing a theorem, formula, equation, or algorithm in domains such as physics, economics, finance, computer science, and math? Answer YES or NO directly.

`{question_and_answer}`

Figure 7: The prompt for determining whether the question is theorem-based.

Explain the answer to the question in a clear and detailed manner. Include citation web links to support the explanation — use relevant Wikipedia pages whenever possible. If a Wikipedia page is not available, use other reliable sources.

`{question_and_answer}`

Figure 8: The prompt for searching relevant web pages using GPT-Search.

Question:

`{question_and_answer}`

Document:

`{document}`

You are a document analysis assistant. Your task is to determine whether the given document above answers the given question and supports the given answer.
**Instructions:**

1. If the answer can be derived or inferred from the text and images in the document, respond YES; otherwise, respond NO.

2. If the document discusses related topics but does not directly answer the given question, respond NO.

3. If the document only provides reference paper titles without substantive content that supports the answer, respond NO.

First, think step by step and explain your reasoning. In the last separate line, directly respond YES or NO without quotes.

Figure 9: The prompt for judging whether the document is relevant to the question.

You are given a markdown document. Your task is to extract the specific theorem, formula, equation, algorithm, or concept named "{theorem_name}" from this document.

**Instructions:**

1. Carefully locate the section that describes the theorem "{theorem_name}".

2. Extract the complete definition, explanation, and any associated formulas or equations.

3. Remove all reference citations.

4. If there are referenced images in the content, preserve the image references exactly as they appear.

5. Your response MUST follow the following LaTeX-style format:

```
\begin{definition}[{theorem_name}]
Complete definition and explanation,
preserving mathematical notation.
Include examples if present.
\end{definition}
```

Here is the document content:
{markdown_content}

Figure 10: The prompt for cleaning the theorem content.

## E  EXPERIMENT DETAILS

### E.1  MODELS AND INSTRUCTIONS

Table 5: Details of the multimodal retriever models evaluated in MRMR.

| Model | Size | Version |
|---|---|---|
| BGE-M3 (Chen et al., 2024) | 600M | BAAI/bge-m3 |
| NE-Embed-V2 (Lee et al., 2025) | 8B | nvidia/NV-Embed-v2 |
| Qwen3-Embedding (Zhang et al., 2025d) | 8B | Qwen/Qwen3-Embedding-8B |
| EVA-CLIP (Sun et al., 2023) | 400M | QuanSun/EVA02-CLIP-L-14 |
| SigLIP (Zhai et al., 2023) | 650M | google/siglip-large-patch16-256 |
| JinaCLIP (Koukounas et al., 2024) | 860M | jinaai/jina-clip-v2 |
| OpenCLIP (Cherti et al., 2023) | 1.4B | laion/CLIP-ViT-g-14-laion2B-s34B-b88K |
| VISTA (Zhou et al., 2024) | 200M | BAAI/bge-visualized-m3 |
| VLM2Vec (Jiang et al., 2025c) | 4B | TIGER-Lab/VLM2Vec-Full |
| GME-Qwen2-VL (Zhang et al., 2025c) | 7B | Alibaba-NLP/gme-Qwen2-VL-7B-Instruct |
| Ops-MM-Embedding (OpenSearch-AI, 2025) | 7B | OpenSearch-AI/Ops-MM-embedding-v1-7B |
| E5-V (Jiang et al., 2025b) | 8B | royokong/e5-v |
| MM-Embed (Lin et al., 2025) | 8B | nvidia/MM-Embed |
| ColPali (Faysse et al., 2025) | 3B | vidore/colpali-v1.3 |

Following TIIR, we evaluate text retrievers on multimodal retrieval tasks by replacing images with captions generated by an LLM. To simulate real-time inference, we apply the standardized prompt "Describe the image" and use Qwen2-VL-2B-Instruct to produce the captions.

### E.2  IMPLEMENTATIONS AND MACHINES

The MRMR dataset is constructed following the conventions of MTEB (Muennighoff et al., 2023), including data format and evaluation pipeline, with modifications to support mixed-modality inputs during evaluation. All experiments are conducted on NVIDIA A100, A6000, or H100 GPUs. The runtime of a full evaluation depends on the model, but with the limited corpus size for efficiency, one complete run can be completed within 4 hours on a single A100 GPU for open-source dense models. To further accelerate dense model evaluation, we employ FlashAttention (Dao et al., 2022).

Table 6: Instruction prompts used during model evaluation in MRMR.

| Task | Modality | Prompt |
|---|---|---|
| Knowledge | Multimodal
Text | Retrieve relevant documents that help answer the question. |
| Theorem | Multimodal
Text | Retrieve relevant theorems that are involved in solving the problem. |
| Negation | Multimodal
Text | Given an image, retrieve descriptions that have contradictory information with the image.
Given an image caption, retrieve descriptions that have contradictory information with the image caption. |
| Vehicle Design | Multimodal
Text | Given a vehicle design, retrieve the design requirements that it violates.
Given a vehicle design description, retrieve the design requirements that it violates. |
| Traffic Case | Multimodal
Text | Given a traffic case, retrieve the driving rule documents that it violates.
Given a traffic case description, retrieve the driving rule documents that it violates. |

## E.3 DETAILED RESULTS

Table 7: Detailed performance of retrieval models on MRMR (*Knowledge*).

| Model | Music | Design | Theo. | Art | Hist. | Soci. | Psy. | Lit. | Pharm. | Diag. | Clinic. | Basic. | Agri. | Geo. | Chem. | Bio. | Avg. |
|---|---|---|---|---|---|---|---|---|---|---|---|---|---|---|---|---|---|
| *Text Models with Image Caption* | | | | | | | | | | | | | | | | | |
| BGE-M3 | 43.4 | 44.0 | 49.4 | 57.2 | 47.7 | 39.5 | 52.2 | 15.8 | 58.5 | 11.2 | 28.2 | 36.2 | 38.7 | 48.6 | 37.6 | 48.3 | 41.0 |
| NV-Embed-v2 | 63.8 | 61.8 | 70.1 | 86.8 | 70.6 | 64.3 | 59.7 | 95.8 | 78.0 | 19.8 | 46.0 | 59.0 | 65.3 | 63.3 | 70.0 | 63.6 | 64.9 |
| Qwen3-Embedding | 62.8 | 62.1 | 74.8 | 87.3 | 76.1 | 74.0 | 69.3 | 97.8 | 83.1 | 34.8 | 47.0 | 64.0 | 69.5 | 76.5 | 74.0 | 72.6 | 70.4 |
| *Text and Image Two-Stream Models with Vector Fusion* | | | | | | | | | | | | | | | | | |
| EVA-CLIP | 30.5 | 1.5 | 3.5 | 7.5 | 16.7 | 5.5 | 16.3 | 0.0 | 22.7 | 10.3 | 10.0 | 16.4 | 41.6 | 15.4 | 20.4 | 18.5 | 14.8 |
| SigLIP | 25.0 | 25.6 | 26.2 | 30.0 | 16.7 | 1.4 | 14.7 | 22.7 | 13.8 | 9.7 | 15.6 | 19.6 | 30.2 | 18.3 | 26.7 | 27.3 | 20.2 |
| OpenCLIP | 20.9 | 50.7 | 62.9 | 86.4 | 35.8 | 10.2 | 15.1 | 22.7 | 11.1 | 10.6 | 20.8 | 25.8 | 34.1 | 45.8 | 23.9 | 34.3 | 31.9 |
| JinaCLIP | 18.5 | 11.0 | 23.0 | 33.1 | 14.2 | 0.0 | 17.1 | 0.0 | 17.8 | 6.1 | 21.7 | 21.1 | 35.1 | 24.7 | 30.4 | 15.4 | 18.1 |
| *Multimodal Models with Merged Image* | | | | | | | | | | | | | | | | | |
| VISTA | 39.3 | 3.5 | 17.2 | 27.5 | 12.3 | 13.9 | 28.0 | 0.0 | 48.9 | 18.2 | 23.9 | 31.2 | 33.6 | 22.0 | 36.9 | 33.1 | 24.3 |
| E5-V | 13.0 | 23.4 | 17.6 | 46.1 | 15.6 | 4.3 | 10.8 | 7.7 | 12.5 | 7.1 | 13.5 | 13.7 | 18.3 | 13.1 | 23.3 | 10.0 | 15.6 |
| MM-Embed | 51.6 | 60.8 | 68.3 | 80.5 | 57.5 | 69.4 | 59.5 | 94.1 | 63.8 | 35.1 | 50.9 | 68.9 | 60.9 | 76.0 | 62.1 | 60.7 | 63.8 |
| VLM2Vec | 34.4 | 44.0 | 49.6 | 84.8 | 36.4 | 12.3 | 19.3 | 19.2 | 17.4 | 13.6 | 23.7 | 33.1 | 39.0 | 40.7 | 37.5 | 30.8 | 33.5 |
| GME-Qwen2-VL | 55.1 | 40.4 | 57.1 | 64.8 | 39.2 | 50.6 | 51.1 | 32.9 | 57.2 | 20.6 | 32.1 | 62.2 | 38.9 | 48.4 | 63.6 | 39.6 | 47.1 |
| Ops-MM-Embedding | 58.5 | 75.6 | 84.2 | 96.8 | 71.4 | 71.1 | 59.7 | 73.7 | 76.1 | 30.9 | 50.7 | 64.5 | 58.7 | 78.5 | 80.4 | 69.0 | 68.7 |
| *Multimodal Models with Document as Image* | | | | | | | | | | | | | | | | | |
| GME-Qwen2-VL | 58.2 | 46.5 | 53.6 | 58.4 | 52.5 | 48.5 | 48.2 | 52.1 | 72.9 | 16.8 | 31.7 | 40.2 | 49.7 | 69.0 | 53.8 | 45.4 | 49.8 |
| Ops-MM-Embedding | 60.6 | 59.0 | 68.4 | 82.4 | 68.3 | 63.0 | 58.6 | 68.3 | 74.3 | 31.2 | 39.3 | 65.9 | 57.2 | 69.3 | 76.1 | 71.9 | 63.4 |
| ColPali | 25.1 | 27.7 | 46.4 | 43.7 | 31.7 | 19.4 | 38.5 | 0.0 | 64.1 | 10.6 | 23.0 | 60.1 | 36.7 | 32.6 | 67.6 | 56.3 | 36.5 |

# F ANALYSIS DETAILS

## F.1 FALSE POSITIVE AND FALSE NEGATIVE ANALYSIS

Although queries and their relevant documents were carefully validated by human annotators, false positives and false negatives may still arise when aggregating documents across queries or sampling from external corpora. We explicitly instructed annotators and validators to identify similar or related queries and to cross-annotate documents accordingly. As a result, some queries share the same relevant documents.

To quantitatively assess the prevalence of such labeling errors, we conducted a human audit of the top-retrieved documents retrieved by the best-performing model Ops-MM-Embedding. As shown in Table 8, the audit revealed zero false positives and a false negative rate of only 2.5% for *Knowledge* tasks for sampled 120 documents. Similarly, for *Theorem* tasks, the combined error rate was minimal at approximately 5.8%, comprising 3.3% false negatives and 2.5% false positives. These results suggest that label noise is insignificant, thereby supporting the reliability of the benchmark.

For *Contradiction* tasks, the dataset is relatively small and predominantly constructed manually by annotators and validators. Given the quality control in the construction process, no additional human evaluation was deemed necessary.

Table 8: Human audit of document relevance annotations for the top-retrieved documents produced by the best-performing multimodal model, Ops-MM-Embedding. A *false negative* is a relevant document incorrectly labeled as irrelevant by our method, and a *false positive* is an irrelevant document incorrectly labeled as relevant.

| | | False Negatives | | False Positives | |
| --- | --- | --- | --- | --- | --- |
| Dataset | Documents Checked | Count | Ratio | Count | Ratio |
| *Theorem* | 120 | 4 | 3.3% | 3 | 2.5% |
| *Knowledge* | 120 | 3 | 2.5% | 0 | 0.0% |

## F.2 ERROR CASE STUDIES

In this section, we present case studies for the Ops-MM-Embedding model in different domains such as Biology (Figure 11) and Traffic Case (Figure 12). The error case analysis for *Theorem* tasks are exemplified as follows.

A recurring issue in Engineering and Geometry tasks is the model's tendency to perform coarse-grained matching based on shape and keywords, while ignoring the specific geometric conditions or physical constraints defined in the query.

**Case Study: Engineering (Geometry)**
*Query ID: validation_Architecture_and_Engineering_5*

- **Query Content:** An image showing a pentagon with internal angles and a specific coordinate bearing angle ($\alpha_{12} = 30°$), asking to calculate other bearing angles.

- **Retrieved Negative (Top-1):** "Inscribing Circle in Regular Pentagon".

- **Analysis:** This represents a **keyword and shape hallucination**. The model correctly identifies the visual object (a pentagon) and the domain (geometry/angles). However, it retrieves a document about inscribing circles—likely because the dense geometric keywords and the visual of a polygon strongly correlate in the embedding space. The model fails to attend to the specific logical task (calculating bearing angles) and instead prioritizes the dominant visual features (the pentagon shape).

**Case Study: Engineering (Statics)**
*Query ID: test_Architecture_and_Engineering_214*

- **Query Content:** A floor plan asking to compute the "tributary areas" for a specific floor beam B1.
- **Retrieved Negative (Top-1):** "Static equilibrium".
- **Analysis:** The retrieved document discusses the static equilibrium of beams. While semantically related to the domain (structural engineering and beams), it is a conceptual mismatch. The model retrieves a theoretical concept (statical indeterminacy) rather than the procedural knowledge required for area calculation. This suggests that the retriever struggles to distinguish between *theoretical concepts* and *practical calculation tasks* when visual cues (schematic diagrams of beams) are similar.

In scientific domains, the model often exhibits "partial understanding", where it correctly identifies the strict sub-domain or topic but fails to retrieve the document addressing the specific variable or relationship queried.

**Case Study: Physics (Thermodynamics)**
*Query ID: test_Physics_74*

- **Query Content:** A $P - V$ (Pressure-Volume) graph showing a cyclic process, asking to identify the point of highest temperature.
- **Retrieved Negative (Top-1):** "Isothermal process".
- **Analysis:** The retrieved document explains isothermal processes (constant temperature), which frequently utilize $P - V$ diagrams similar to the query image. The retrieval is plausible but incorrect; the model latched onto the visual graph type ($P - V$ curve) but failed to deduce the specific relationship ($PV = nRT$) required to find the maximum temperature. This confirms the limitation regarding **higher-level deduction**: the model recognizes the graphical language but not the specific physical implication.

**Case Study: Math (Data Interpretation)**
*Query ID: test_Math_469*

- **Query Content:** A histogram/bar chart showing student distances from school, asking for the percentage of students whose distance falls within the 5 km to 10 km range.
- **Retrieved Negative (Top-1):** "Generic statistical methods".
- **Analysis:** The retrieved document discusses general statistical techniques and data representation, which often involve histograms similar to the query image. The retrieval appears contextually related due to the presence of a histogram but is ultimately incorrect; the model associated the visual format with a broad statistical category but failed to extract or interpret the specific numerical data (bin counts and ranges) needed to compute the required percentage. This highlights a deficiency in **visual-numerical alignment**: the model recognizes the chart type but does not connect it to the mathmatical reasoning.

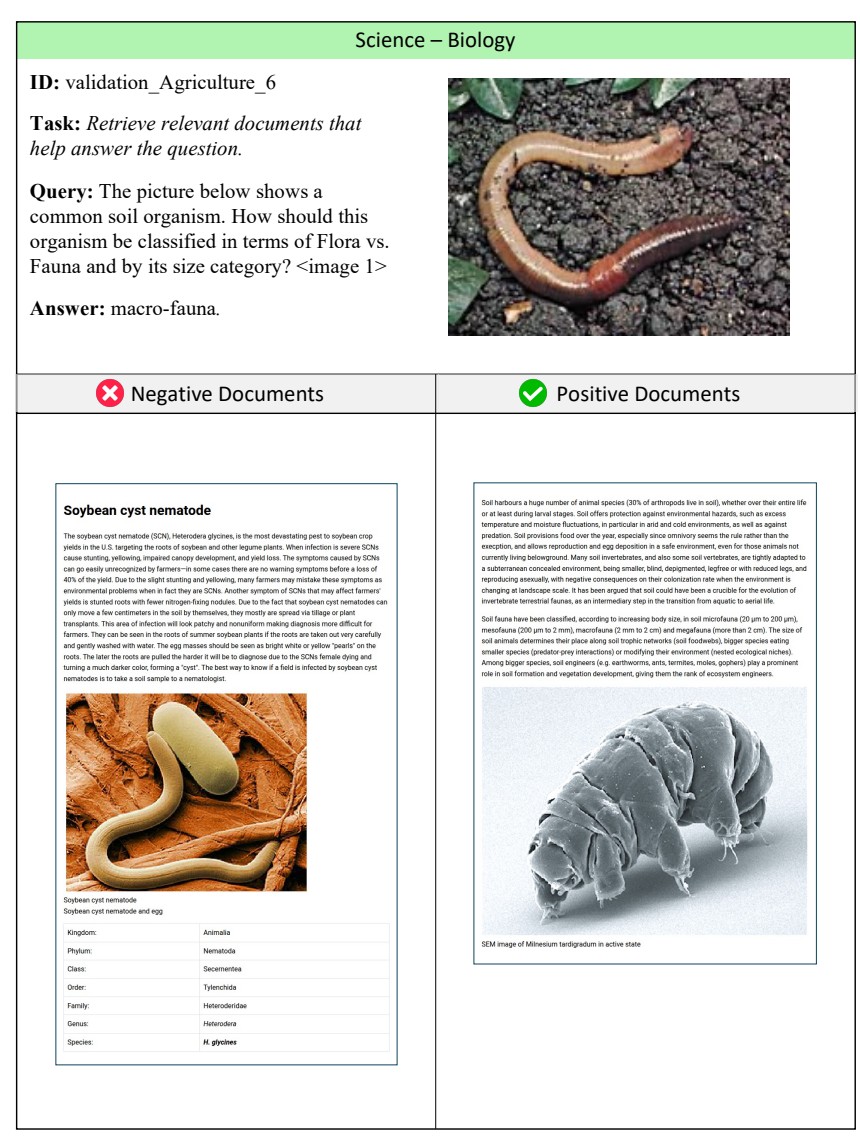

Figure 11: Error case example in Agriculture where the multimodal embedding model Ops-MM-Embedding prioritizes the negative document in the left over the positive document in the right.

 Back to Appendix Table of Contents

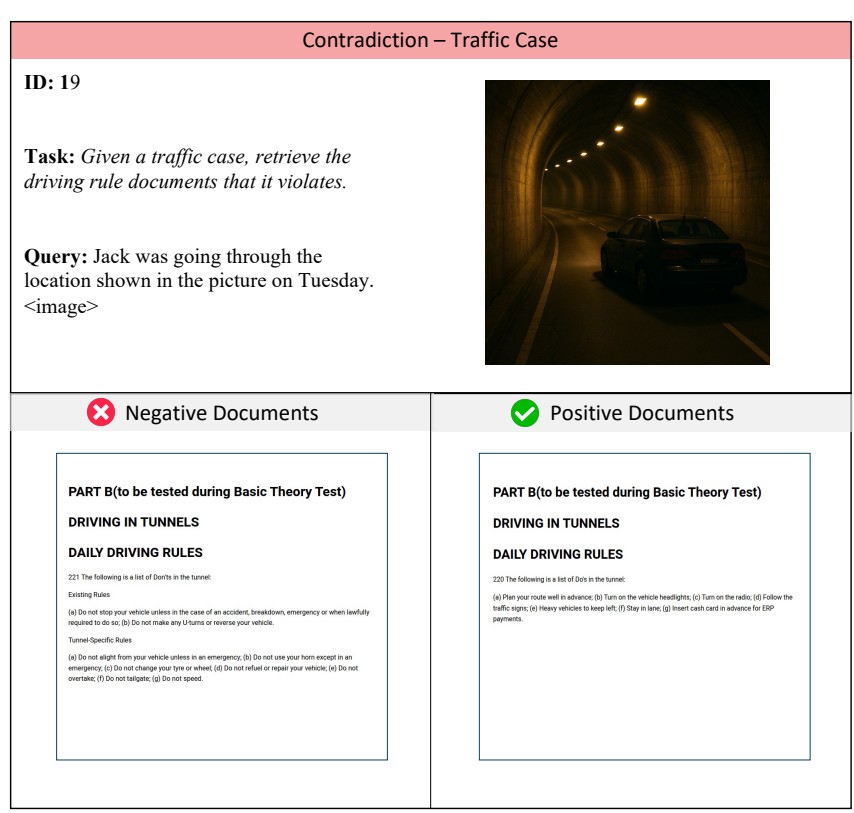

Figure 12: Error case example in Traffic where the multimodal embedding model Ops-MM-Embedding prioritizes the negative document in the left over the positive document in the right.

     Back to Appendix Table of Contents

We conducted query expansion experiments using both weak and strong vision-language models (VLMs)—namely, Qwen2-VL-2B and Qwen2.5-VL-72B—for weak and strong multimodal retrievers (i.e., GME-Qwen2-VL and Ops-MM-Embedding). As shown in Tables 9 and 10, query expansion is generally effective for weak retriever models. However, for stronger retrievers, the quality of the query expansion becomes critical: expansions generated by the weaker VLM actually degrade the performance of the stronger retriever.

With query expansion by a strong LLM (Qwen2.5-VL-72B), the expansion technique is effective for improving both strong and weak retriever models. However, they are still far from perfect on this benchmark. For example, the best retriever with strong query expansion only achieves 55.9 for medical queries and 31.8 for math queries.

Table 9: nDCG@10 scores of the multimodal retriever GME-Qwen2-VL on MRMR *Knowledge* and *Theorem* tasks, comparing the original queries with query expansions generated by Qwen2-VL-2B-Instruct and Qwen2.5-VL-72B-Instruct. The average query length ($Q$ #Text) before and after expansion is reported as the number of tokens measured by the GPT-2 tokenizer.

| Model | Knowledge | | | | | Theorem | | | | | Avg. |
|---|---|---|---|---|---|---|---|---|---|---|---|
| | $Q$ #Text | Art | Med. | Sci. | Hum. | $Q$ #Text | Math | Phy. | Eng. | Bus. | |
| Original | 31.4 | 54.3 | 40.1 | 46.8 | 45.6 | 58.6 | 28.8 | 36.0 | 30.2 | 45.1 | 40.9 |
| Qwen2-VL-2B | 699.6 | 64.9 | 49.6 | 64.6 | 48.9 | 735.9 | 23.5 | 36.5 | 31.5 | 48.3 | 46.0 |
| Qwen2.5-VL-72B | 843.8 | 76.9 | 61.8 | 77.0 | 72.2 | 1218.4 | 33.3 | 36.9 | 32.7 | 55.0 | 55.7 |

Table 10: nDCG@10 scores of the multimodal retriever Ops-MM-Embedding on MRMR *Knowledge* and *Theorem* tasks, comparing the original queries with query expansions generated by Qwen2-VL-2B-Instruct and Qwen2.5-VL-72B-Instruct. The average query length ($Q$ #Text) before and after expansion is reported as the number of tokens measured by the GPT-2 tokenizer.

| Model | Knowledge | | | | | Theorem | | | | | Avg. |
|---|---|---|---|---|---|---|---|---|---|---|---|
| | $Q$ #Text | Art | Med. | Sci. | Hum. | $Q$ #Text | Math | Phy. | Eng. | Bus. | |
| Original | 31.4 | 79.3 | 52.5 | 70.0 | 67.8 | 58.6 | 27.7 | 39.5 | 30.1 | 52.3 | 52.4 |
| Qwen2-VL-2B | 699.6 | 77.2 | 45.7 | 67.1 | 58.0 | 735.9 | 24.7 | 35.6 | 29.5 | 50.2 | 48.5 |
| Qwen2.5-VL-72B | 843.8 | 80.5 | 55.9 | 73.1 | 64.6 | 1218.4 | 31.8 | 39.4 | 35.3 | 53.9 | 54.3 |

## G Data Examples

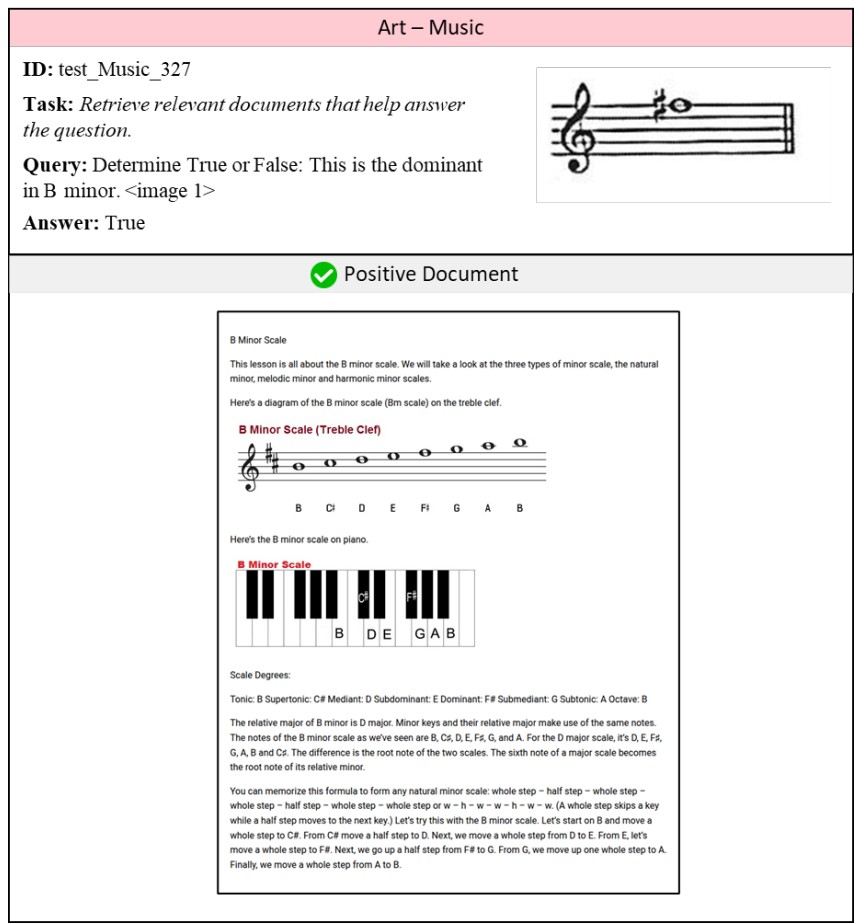

Figure 13: Music example.

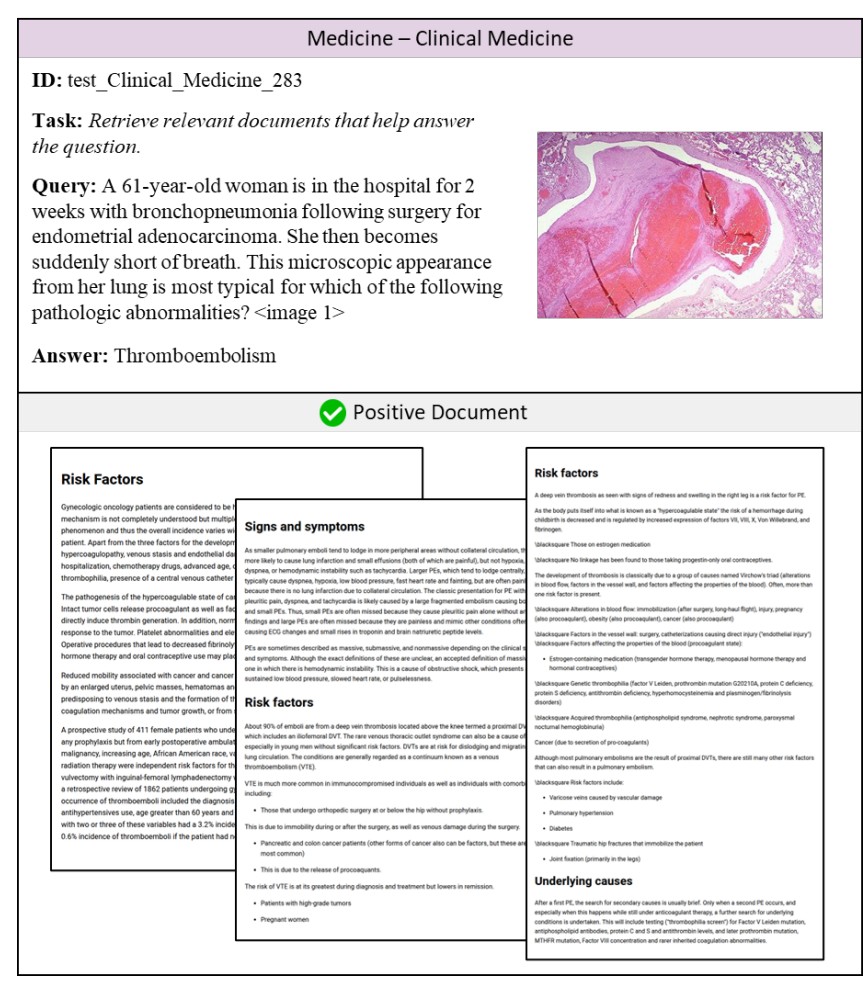

Figure 14: Clinic Medicine example.

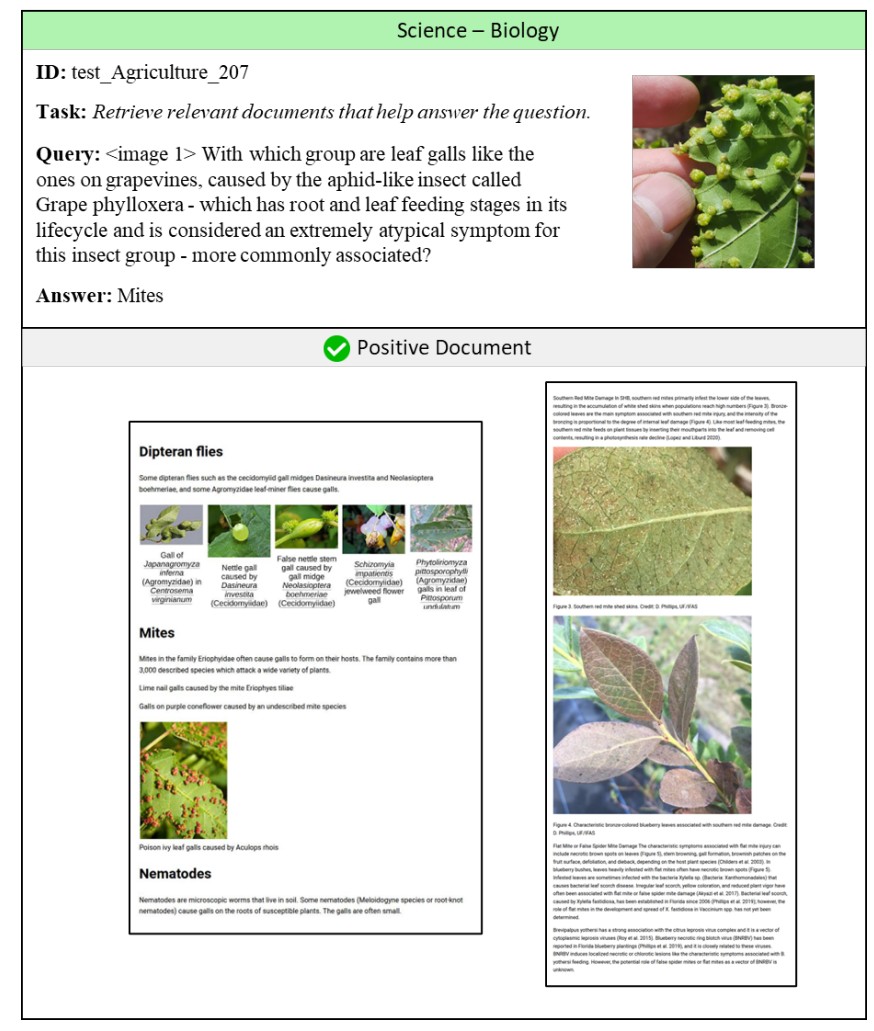

Figure 15: Biology example.

**Humanities – Psychology**

**ID:** validation_Sociology_1

**Task:** *Retrieve relevant documents that help answer the question.*

**Query:** In 1946, the person in <image 1> was arrested for refusing to sit in the blacks-only section of the cinema in Nova Scotia. This is an example of_______________.

**Answer:** A conflict crime

✅ Positive Document

**Conflict criminology**

Largely based on the writings of Karl Marx, conflict criminology holds that crime in capitalist societies cannot be adequately understood without a recognition that such societies are dominated by a wealthy elite whose continuing dominance requires the economic exploitation of others, and that the ideas, institutions and practices of such societies are designed and managed in order to ensure that such groups remain marginalised, oppressed and vulnerable. Members of marginalised and oppressed groups may sometimes turn to crime in order to gain the material wealth that apparently brings equality in capitalist societies, or simply in order to survive. Conflict criminology derives its name from the fact that theorists within the area believe that there is no consensual social contract between state and citizen.

**Discussion**

Conflict theory assumes that every society is subjected to a process of continuous change and that this process creates social conflicts. Hence, social change and social conflict are ubiquitous. Individuals and social classes, each with distinctive interests, represent the constituent elements of a society. As such, they are individually and collectively participants in this process but there is no guarantee that the interests of each class will coincide. Indeed, the lack of common ground is likely to bring them into conflict with each other. From time to time, each element's contribution may be positive or negative, constructive or destructive. To that extent, therefore, the progress made by each society as a whole is limited by the acts and omissions of some of its members by others. This limitation may promote a struggle for greater progress but, if the less progressive group has access to the coercive power of law, it may entrench inequality and oppress those deemed less equal. In turn, this inequality will become a significant source of conflict. The theory identifies the state and the law as instruments of oppression used by the ruling class for their own benefit.

There are various strands of conflict theory, with many heavily critiquing the others. Structural Marxist criminology, which is essentially the most 'pure' version of the above, has been frequently accused of idealism, and many critics point to the fact that the Soviet Union and such states had as high crime rates as the capitalist West. Furthermore, some highly capitalist states such as Switzerland have very low crime rates, thus making structural theory seem improbable.

Figure 16: Psychology example.

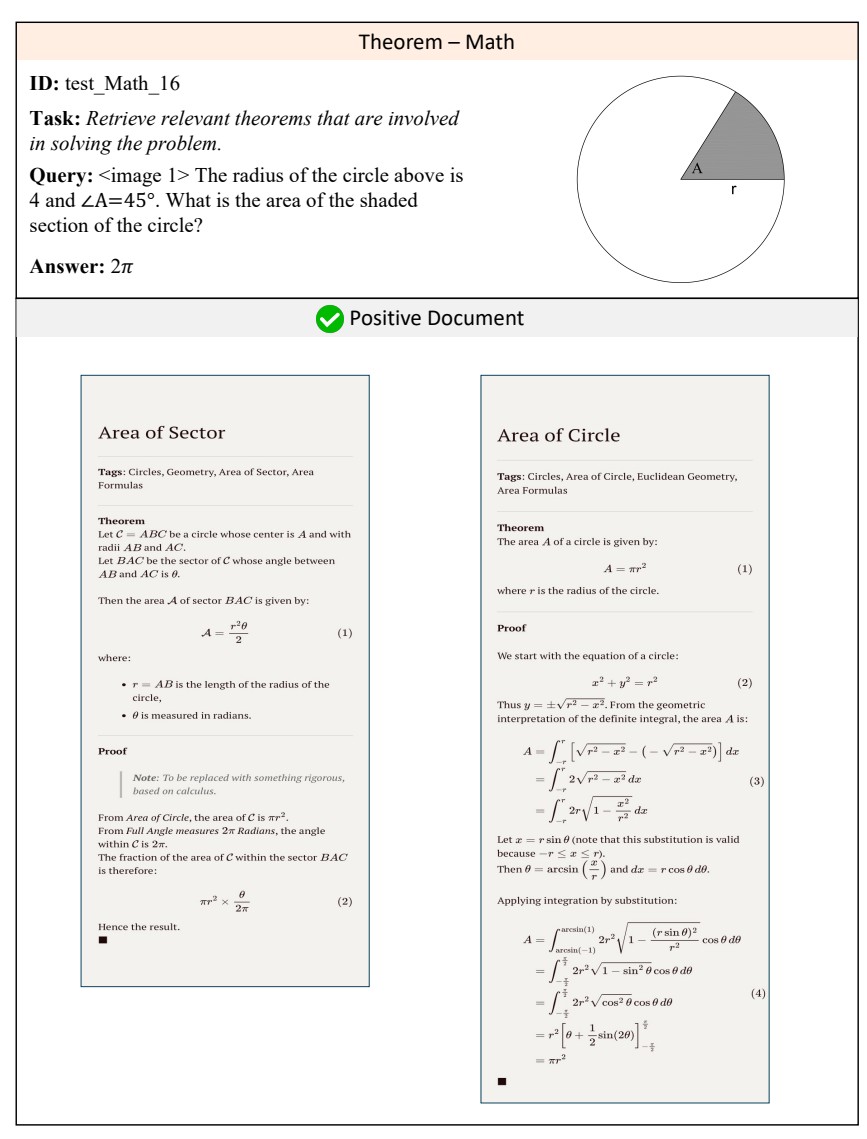

Figure 17: Math example.

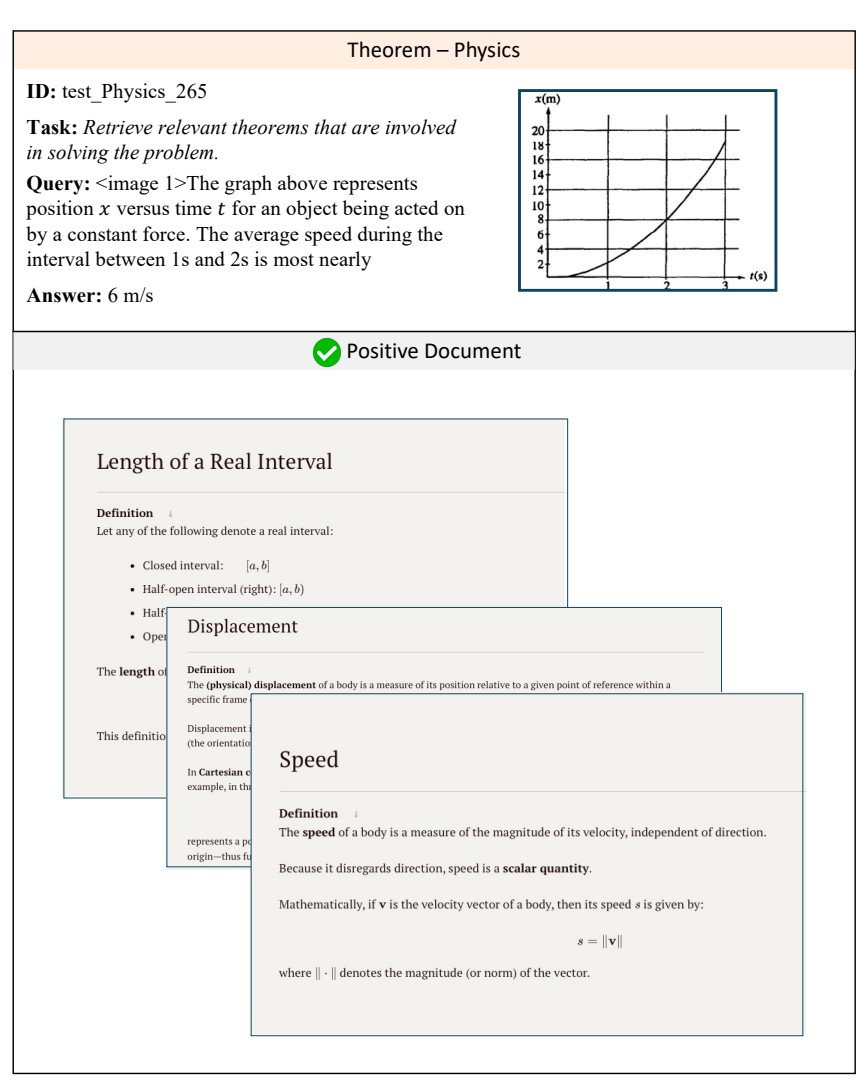

Figure 18: Physics example.

 

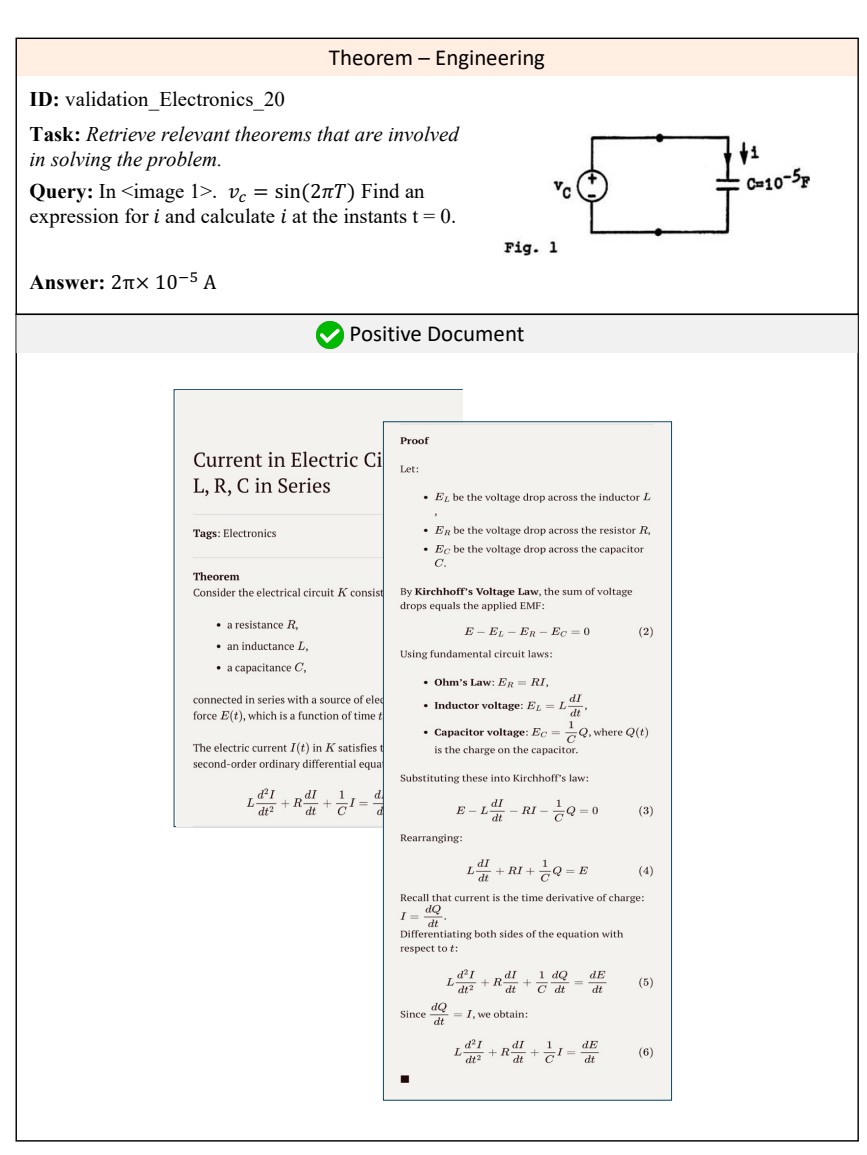

Figure 19: Engineering example.

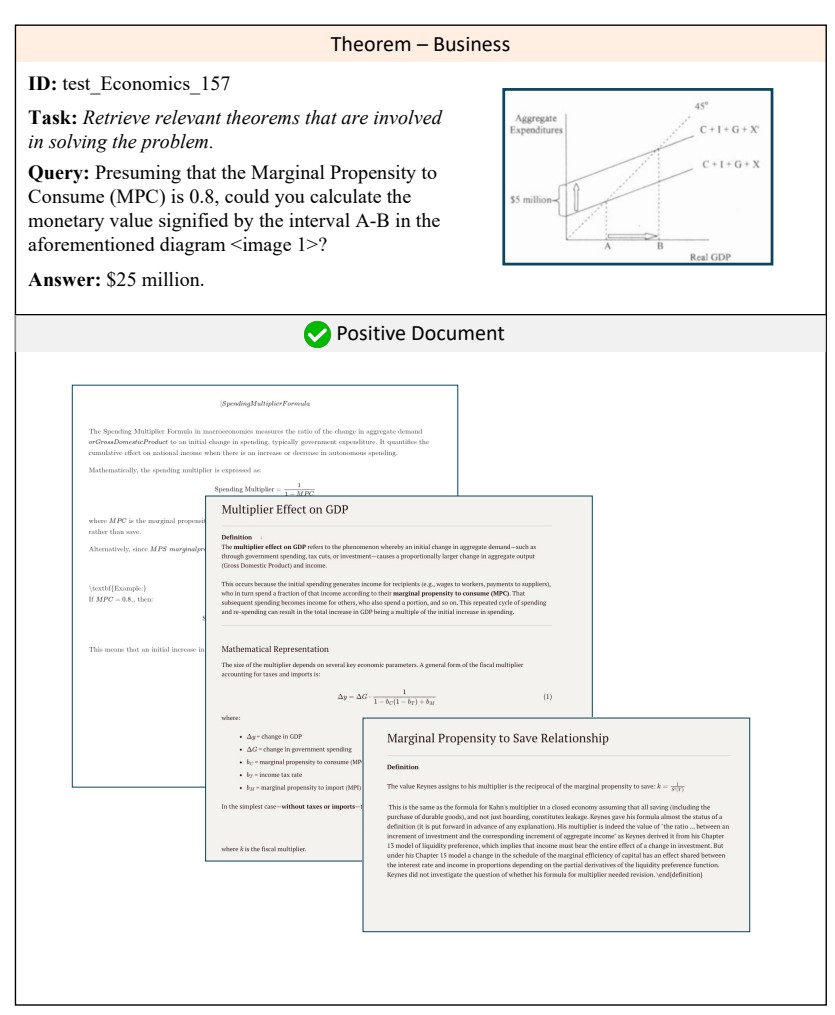

Figure 20: Business example.

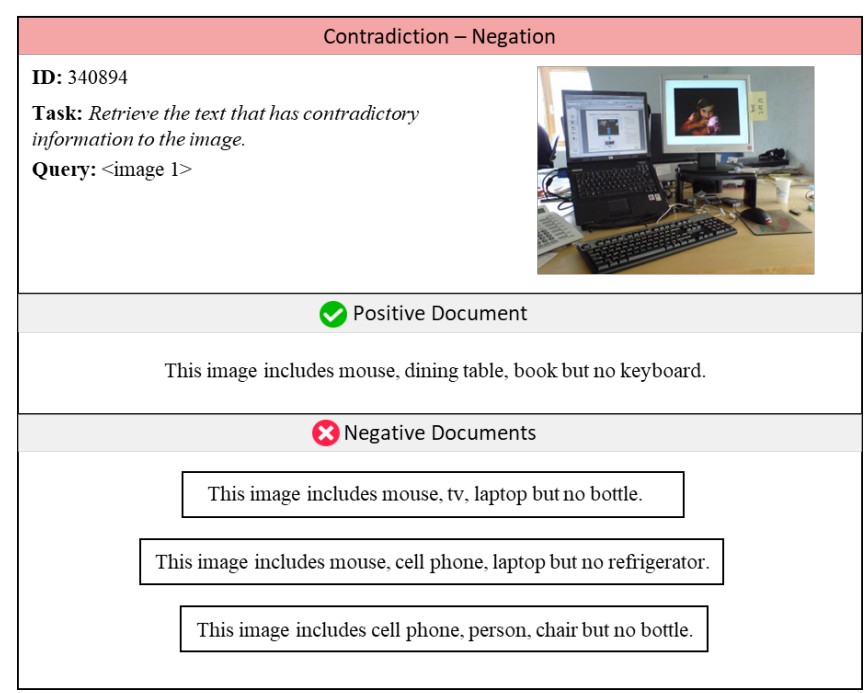

Figure 21: Negation example.

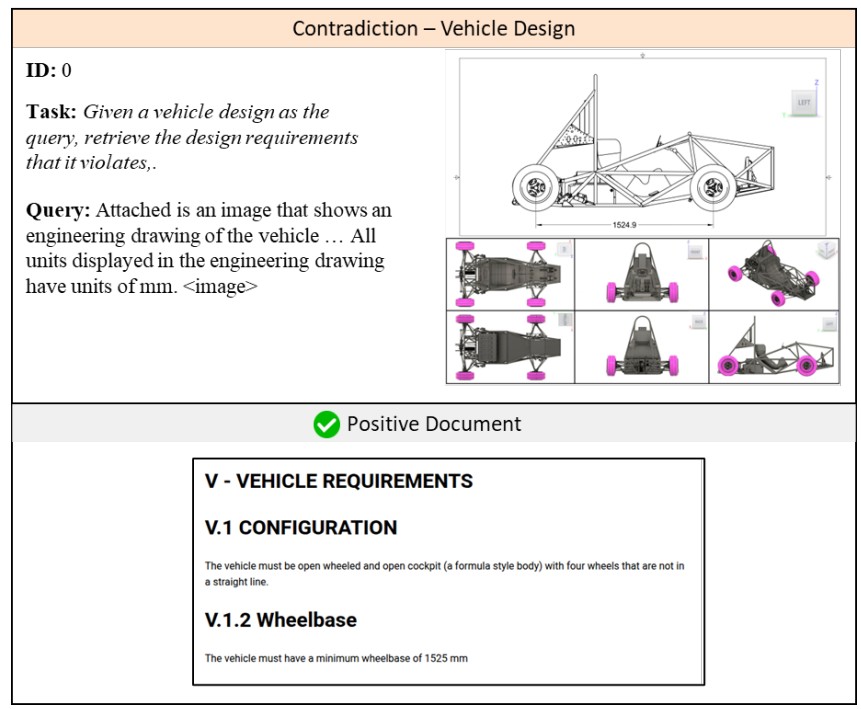

Figure 22: Vehicle Design example.

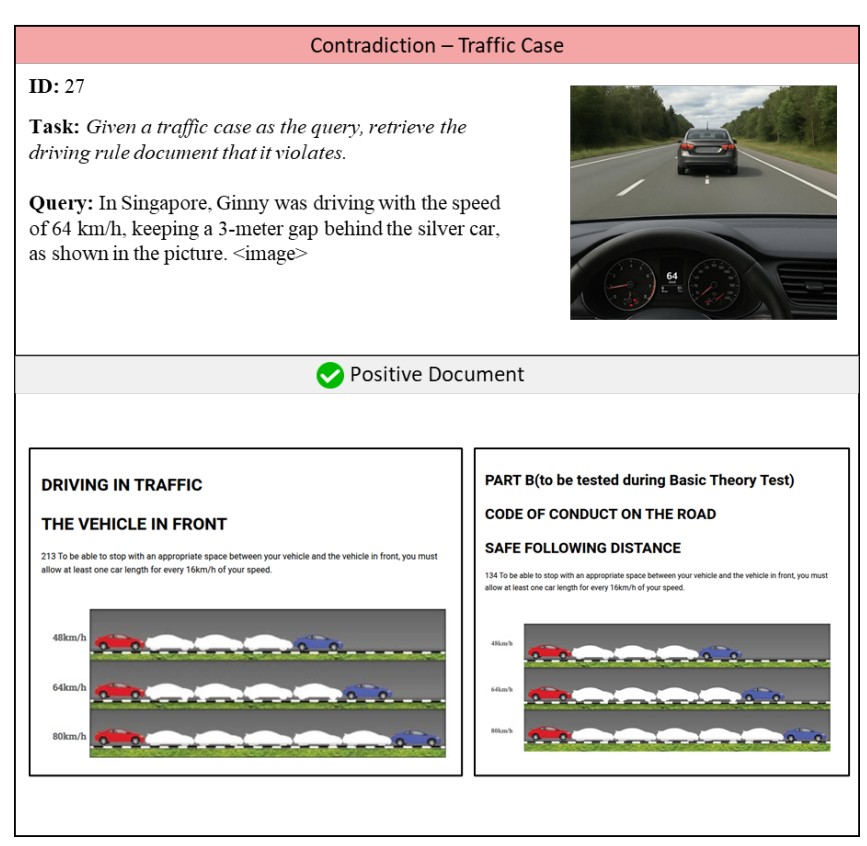

Figure 23: Traffic Case example.

        Back to Appendix Table of Contents