# OpenReview forum: "MRMR: A Realistic and Expert-Level Multidisciplinary Benchmark for Reasoning-Intensive Multimodal Retrieval"
_ICLR.cc/2026/Conference — ICLR 2026 Poster_

### Official Review · Reviewer_9XX3 · 2025-10-22

**Soundness:** 4
**Presentation:** 3
**Contribution:** 2
**Rating:** 6
**Confidence:** 4

**Summary:**

This paper introduces a multimodal retrieval benchmark with an emphasis on reasoning-intensive retrieval across 23 domains.
In the proposed benchmark, the authors introduce three distinct types of retrieval tasks -- knowledge retrieval, theorem retrieval, and contradiction retrieval.
The paper evaluates various multimodal retrieval setups (e.g., text-only, two-stream, multimodal, and document-as-image models) covering several frontier models on the MRMR benchmark. The results suggest that existing multimodal retrievers struggle on reasoning-intensive and contradiction retrieval tasks compared to text-only retrievers.
Overall, this paper presents a meaningful contribution to the community, though its technical contribution is limited.

**Strengths:**

1. Wide domain coverage: MRMR spans 23 domains across six disciplines.
2. Integration of contradiction retrieval: adding the negation/contradiction retrieval testbed is practically useful and can be a good initiative for the community.
3. Comprehensive evaluation: the experiments cover a wide range of retrieval architectures and models.
Human-verified annotation and corpus quality control: All positive documents are validated by human experts, and the corpus is cleaned under human supervision, making MRMR more realistic compared to fully synthetic benchmarks.

**Weaknesses:**

1. Limited technical contribution beyond task composition: it's unclear about the main technical contribution beyond composing tasks (e.g., reasoning-intensive, image-text interleaving, multidisciplinary) from prior works.
2. Limited dataset size: as a benchmark spanning 23 domains, size of the dataset (1502 queries) is relatively small and therefore may limit statistical robustness.
3. Writing clarity: Certain sections could be written more clearly to improve readability and logical flow, particularly in the methodology and evaluation descriptions.

**Questions:**

1. In table 2, the third last row (L181), what does it mean that negation task has 4 documents in total (D=4), shouldn't it be 200*4=800?
2. L220, should "solvable" be "unsolvable"?
3. Can you discuss similar works such as $\text{MR}^2$ bench [1] on how MRMR differentiates from them?
4. Can you add prompts in the appendix that you use to synthesize the dataset (e.g., synthesize negation candidates for the contradiction retrieval task)?

[1] Zhou, Junjie, et al. "MR $^ 2$-Bench: Going Beyond Matching to Reasoning in Multimodal Retrieval." arXiv preprint arXiv:2509.26378 (2025).

---

> ### Author Response · Authors · 2025-11-20
> **Questions**
>
> Thank you very much for your careful reading and thoughtful feedback! We’re pleased that the reviewer found the benchmark diverse, the experiments comprehensive, and the data collection pipeline rigorous. Below, we address the concerns raised.
>
> > 1. In table 2, the third last row (L181), what does it mean that negation task has 4 documents in total (D=4), shouldn't it be 200*4=800?
>
> Thank you for the comment. For this Negation subset, we adopted a **multi-choice setup** similar to the BLINK task in the MIEB benchmark [1]. Unlike each query of Knowledge and Theorem that has all documents as candidates, each Negation query only has 4 text documents as candidates. There is no shared corpus for Negation queries. We have shown that even with such a small number of candidates, the models fail to find conflicting statements about object existence. L181 shows the number of candidate documents for each query.
>
> > 2. L220, should "solvable" be "unsolvable"?
>
> Thanks for the question. To ensure that the image is essential for answering each multimodal question, MMMU-Pro validates the dataset by having a text-only LLM attempt to answer the questions. If the text-only model can answer correctly, the question is considered **“solvable” without the image and is removed from the dataset**. This process ensures that the remaining questions genuinely require visual information. This is one of the key reasons we selected MMMU-Pro for our benchmark.
>
> > 3. Can you discuss similar works such as  bench [1] on how MRMR differentiates from them?
>
> Thank you for mentioning the contemporaneous ICLR 2026 submission MR2-Bench [2]. MRMR and MR2-Bench share similar goals in advancing reasoning-intensive and text–image interleaved multimodal retrieval, highlighting the need for such benchmarks in this area. Uniquely, MRMR covers a much broader range of disciplines, including high-stakes domains such as Pharmacy, Diagnostics, Clinical Medicine, Agriculture, and Music, and it provides human-expert relevance annotations. More importantly, MRMR introduces an entirely new task, i.e., Contradiction Retrieval, to evaluate models' ability to handle conflicting concepts. This task is constructed from scratch with newly created queries and documents, rather than simply aggregating existing datasets.
>
> > 4. Can you add prompts in the appendix that you use to synthesize the dataset (e.g., synthesize negation candidates for the contradiction retrieval task)?
>
> Thanks for pointing this out. We have improved the section Appendix D for the dataset construction details in the new version.
>
> As explained in Section D.1, the Negation candidates are generated using predefined templates such as ``The image includes $a$, $b$, $c$, but no $d$``, where each symbol corresponds to a (non)existent object label sampled from COCO annotations. No LLM prompting is involved in constructing the Negation task.
>
> For the Traffic Case and Vehicle Design datasets (Sections D.2 and D.3), we primarily rely on human annotations: annotators wrote the traffic case storylines, and the vehicle design images were manually augmented through human editing. Therefore, these subsets have a limited number of queries.
>
> As for Knowledge and Theorem tasks, we have added Appendix B.4 in the revised version for dataset construction prompts following your suggestion.
>
>
> [1] Xiao, C., Chung, I., Kerboua, I., Stirling, J., Zhang, X., Kardos, M., & Solomatin, R. (2025). MIEB: Massive Image Embedding Benchmark.
>
> [2] Zhou, Junjie, et al. "MR -Bench: Going Beyond Matching to Reasoning in Multimodal Retrieval." arXiv preprint arXiv:2509.26378 (2025).

---

> ### Author Response · Authors · 2025-11-20
> **Weaknesses**
>
> > 1. Limited technical contribution beyond task composition: it's unclear about the main technical contribution beyond composing tasks (e.g., reasoning-intensive, image-text interleaving, multidisciplinary) from prior works.
>
> We thank the reviewer for the feedback. Our contribution goes beyond assembling tasks. MRMR introduces three core technical advances:
>
> (1) **Multidisciplinary expert coverage**: the first multimodal retrieval benchmark spanning 23 expert domains (medicine, engineering, agriculture, business), far beyond prior general-domain datasets. The previous multimodal retrieval tasks involved only general-domain images and simple text queries.
>
> (2) Reasoning-intensive multimodal retrieval: we extend theorem-based and expert-knowledge reasoning to the multimodal setting, revealing failure modes not captured by existing benchmarks.
>
> (3) Interleaved image–text retrieval: MRMR is the first to represent both queries and documents as interleaved sequences with multiple images.
>
> In addition, we introduce a **novel retrieval task**—Contradiction Retrieval—along with a semi-automated pipeline for constructing high-quality interleaved documents. Finally, we present a **comprehensive evaluation** across four multimodal retrieval setups using 14 state-of-the-art models, revealing key limitations of current approaches. These elements together constitute a substantial technical contribution beyond task composition.
>
>
> > 2. Limited dataset size: as a benchmark spanning 23 domains, size of the dataset (1502 queries) is relatively small and therefore may limit statistical robustness.
>
> Thanks for the comment. As shown in Table 1, MRMR contains 1,435 expert-annotated queries across 23 domains. While this may appear small by NLP standards, in the information retrieval (IR) community, benchmarks with 30–50 queries are often considered valid and effective, since each query requires meticulous relevance assessment across many candidate documents [1][2]. This is the primary reason we curate approximately 100 queries for each subtask. Moreover, recent text and multimodal retrieval benchmarks typically contain a comparable number of queries. We summarize them below for reference.
>
> | Benchmark      | Modality                              | #Queries | Task / Focus                                                                 |
> | -------------- | -------------------------------------- | -------- | ----------------------------------------------------------------------------- |
> | FollowIR [1]   | Text                                   | 349      | Re-ranking documents according to user-provided instructions                  |
> | BRIGHT [3]     | Text                                   | 1,398    | Reasoning-intensive text retrieval (math, code, social science, etc.)        |
> | CIRR [4]       | Multimodal                             | 4,148    | Retrieve real-life images that match a text-guided modification of a reference image |
> | ViDoRe [5]     | Multimodal                             | 3,810    | Retrieve multimodal document pages containing text, figures, and tables   |
> | MMDocIR [6]    | Multimodal                             | 1,658    | Retrieve multimodal document pages containing text, figures, and tables           |
> | MR2-Bench [7]  | Multimodal                             | 1,309    | Reasoning-oriented multimodal retrieval                                      |
> | **MRMR (Ours)**| **Multimodal**| **1,502** | **Reasoning-intensive retrieval across Knowledge, Theorem, and Contradiction tasks** |
>
>
> Importantly, MRMR is costly because, like BRIGHT [3] and MR2‑Bench [7], it requires domain experts (e.g., clinicians, engineers) to judge not just topical relevance but whether a document genuinely supports the reasoning needed to answer the query. Each query involves constructing or validating complex multimodal scenarios, sourcing relevant documents from the web, and distinguishing true evidence from hard negatives. This high per‑query cost limits scale but ensures high-quality, reasoning-intensive labels.
>
>
> > 3. Writing clarity: Certain sections could be written more clearly to improve readability and logical flow, particularly in the methodology and evaluation descriptions.
>
> We appreciate the reviewer’s suggestion. We have revised the methodology and evaluation sections for improved clarity and logical flow in the new version, including restructuring the data construction pipeline for smoother step-by-step explanation, clarifying how each retrieval task is formulated, and reorganizing the evaluation setup with clearer categorization. These revisions significantly enhance readability and make the contributions easier to follow.

---

> ### Author Response · Authors · 2025-11-20
> **Weaknesses (References)**
>
> [1] Weller, O., Chang, B., MacAvaney, S., Lo, K., Cohan, A., Van Durme, B., and Soldaini, L. 2024. FollowIR: Evaluating and Teaching Information Retrieval Models to Follow Instructions. Proceedings of the 2025 Conference of the Nations of the Americas Chapter of the Association for Computational Linguistics: Human Language Technologies (Volume 1: Long Papers).
>
> [2] Webber, W., Moffat, A., and Zobel, J. 2008. Statistical power in retrieval experimentation. In Proceedings of the 17th ACM Conference on Information and Knowledge Management (CIKM 2008).
>
> [3] Su, H., Yen, H., Xia, M., Shi, W., Muennighoff, N., Wang, H., Haisu, L., Shi, Q., Siegel, Z. S., Tang, M., Sun, R., Yoon, J., Arik, S. O., Chen, D., and Yu, T. 2025. BRIGHT: A Realistic and Challenging Benchmark for Reasoning-Intensive Retrieval. In Proceedings of the International Conference on Learning Representations (ICLR 2025).
>
> [4] Liu, Z., Rodriguez-Opazo, C., Teney, D., and Gould, S. 2021. Image retrieval on real-life images with pre-trained vision-and-language models. In Proceedings of the IEEE/CVF International Conference on Computer Vision (ICCV 2021).
>
> [5] Macé, Q., Loison, A., and Faysse, M. 2025. ViDoRe Benchmark V2: Raising the Bar for Visual Retrieval. arXiv preprint arXiv:2505.17166.
>
> [6] Dong, K., Chang, Y., Goh, X. D., Li, D., Tang, R., and Liu, Y. 2025. MMDocIR: Benchmarking Multimodal Retrieval for Long Documents. In Proceedings of the 2025 Conference on Empirical Methods in Natural Language Processing (EMNLP 2025).
>
> [7] Anonymous. 2025. MR²-Bench: Going Beyond Matching to Reasoning in Multimodal Retrieval. Submitted to International Conference on Learning Representations (ICLR 2025).

---

### Official Review · Reviewer_8qw3 · 2025-10-30

**Soundness:** 3
**Presentation:** 3
**Contribution:** 3
**Rating:** 8
**Confidence:** 4

**Summary:**

This paper introduces MRMR (Multidisciplinary Reasoning-intensive Multimodal Retrieval), a novel benchmark designed to evaluate retrieval systems in expert-level, reasoning-intensive, and multimodal tasks. Unlike previous benchmarks that focus on general domains, MRMR spans 23 domains, including medicine, engineering, science, and business, and features 1,502 expert-annotated queries. The tasks are categorized into three types: Knowledge Retrieval, Theorem Retrieval, and Contradiction Retrieval. MRMR challenges retrieval models not only to identify semantically relevant documents but also to perform deep reasoning and logical deduction, such as identifying contradictory concepts or retrieving relevant theorems for complex calculations. The dataset includes interleaved image-text sequences, reflecting the real-world scenario where queries and documents are multimodal. The paper evaluates a variety of state-of-the-art multimodal retrieval systems and finds significant room for improvement, particularly in reasoning-intensive tasks.

**Strengths:**

* MRMR provides a unique and comprehensive framework to evaluate multimodal retrieval systems on expert-level, reasoning-intensive tasks across multiple disciplines. It addresses the gap in existing benchmarks that fail to capture the complexity of real-world, domain-specific applications, such as medical diagnoses and engineering design.
* The benchmark spans 23 domains, providing a holistic evaluation of retrieval systems across various expert fields. This diversity helps to assess the models' generalizability and specialization across disciplines like medicine, business, engineering, and the sciences.
* By using interleaved image-text sequences, MRMR more accurately reflects real-world retrieval tasks, where queries and documents are rarely confined to a single modality. This design enhances the practical relevance of the benchmark.

**Weaknesses:**

1. Have the authors considered the scenario where the documents highly relevant to a given query come from multiple different disciplines? For example, the user’s question might require the synthesis of documents from multiple disciplines to provide a comprehensive answer. This kind of situation frequently arises in real-world applications. Did the authors consider this when designing the benchmark?

2. In the overall design of the benchmark, the authors separated documents by discipline for the experiments. However, creating a large database by mixing documents from all disciplines and performing retrieval might present a more challenging setting. This cross-disciplinary retrieval scenario could provide an even more difficult test for the models.

3. The authors discuss retrieval within specific disciplines, which is an interesting topic, but they have not addressed cross-disciplinary retrieval. For instance, if the query includes an image of a sculpture from the art domain and the user wants to retrieve mathematical theorems (from the math domain) related to the geometry of the sculpture, this is a very interesting problem. I look forward to the authors' response and discussion on this question.

**Questions:**

Please refer to the weaknesses.

---

> ### Author Response · Authors · 2025-11-16
>
> Thanks for your thoughtful review, recognition and constructive feedback on our work. We have carefully thought about your comments as follows.
>
> > Have the authors considered the scenario where the documents highly relevant to a given query come from multiple different disciplines? For example, the user’s question might require the synthesis of documents from multiple disciplines to provide a comprehensive answer. This kind of situation frequently arises in real-world applications. Did the authors consider this when designing the benchmark?
>
> That’s a very interesting idea—evaluating models on their ability to retrieve or synthesize knowledge across multiple disciplines. Currently, most MRMR queries require knowledge from only a **single discipline**. It would definitely be worthwhile to explore such a challenging yet realistic benchmark in future work.
>
> > In the overall design of the benchmark, the authors separated documents by discipline for the experiments. However, creating a large database by mixing documents from all disciplines and performing retrieval might present a more challenging setting. This cross-disciplinary retrieval scenario could provide an even more difficult test for the models.
>
> Thank you for the comment. We agree that mixing documents from multiple disciplines creates a more challenging retrieval setting. As shown in Table 2, for the knowledge-intensive retrieval task, we combined documents from Art, Medicine, Science, and Humanities into a shared corpus of 26,223 documents. Since most MRMR queries focus on a single discipline, it was unnecessary to annotate cross-disciplinary relevance; we simply assume that no additional relevant documents exist outside the target discipline. We did the same for the 14,257 theorem-related documents.
>
> For the Knowledge, Theorem, and Contradiction categories, we kept their corpora separate to ensure **evaluation efficiency**. Since theorem documents are highly unlikely to be relevant to Knowledge queries (and vice versa) in our design, using separate corpora significantly reduces unnecessary retrieval and evaluation overhead.
>
> > The authors discuss retrieval within specific disciplines, which is an interesting topic, but they have not addressed cross-disciplinary retrieval. For instance, if the query includes an image of a sculpture from the art domain and the user wants to retrieve mathematical theorems (from the math domain) related to the geometry of the sculpture, this is a very interesting problem. I look forward to the authors' response and discussion on this question.
>
> Thank you for the excellent suggestion. We fully agree that this **cross-disciplinary task** adds an additional layer of complexity for multimodal retrieval models. Our work aims to construct the first multi-disciplinary multimodal retrieval benchmark. Currently, there is no multi-disciplinary benchmarks with image-text interleaved queries. And your idea would certainly merit a dedicated follow-up work/benchmark in the future.

---

### Official Review · Reviewer_8xXt · 2025-10-31

**Soundness:** 3
**Presentation:** 2
**Contribution:** 3
**Rating:** 6
**Confidence:** 4

**Summary:**

Summary:
Authors repurpose MMMU-Pro and use other synthetic methods to generate a multimodal retrival dataset that has expert domain, reasoning intensive and contradition style questions.

**Strengths:**

Strengths:
- Addresses gaps in the present day benchmarks which mostly rely on surface level/minimal reasoning.
- Clear writing on how each datapoint was created.
- Also provide hard negatives which can be useful to train models.

**Weaknesses:**

Weakness:
- Positives of each question are curated independetly. Positives of all documents are put together to construct the corpus. How was it ensured that positives of other queries are not positives of a given query.
  - What were false negatives filtered?
  - Similarly, negatives mined for other queries either by GPT-Search/Human mismatch or by PIN14 can turn out to be potential positves of other queries.
- Smaller dataset compared to sme of the MTEB/MMEB datasets.
- Evaluation only benchmark. It would have been significantly more useful had a training dataset been provided.
- Query expansion results for stronger retriever models missing. It is important to understand if query exapansion with multimodal/text retriver models still falls short on this benchmark.
- Weak qualitative analysis. A category wise qualitative analysis would have been much more helpful.

**Questions:**

-

**Details Of Ethics Concerns:**

-

---

> ### Author Response · Authors · 2025-11-20
> **Weaknesses (1)**
>
> Thank you for taking the time to review our paper and provide valuable feedback. We appreciate your insights and would like to address your concerns as follows.
>
> > 1. Positives of each question are curated independently. Positives of all documents are put together to construct the corpus. How was it ensured that positives of other queries are not positives of a given query. What were false negatives filtered? Similarly, negatives mined for other queries either by GPT-Search/Human mismatch or by PIN14 can turn out to be potential positives of other queries.
>
> Thank you for the thoughtful insights. We have also considered these issues. Since most of the questions and sampled documents from PIN14 are highly diverse, the probability that a randomly selected document is relevant to any given question is quite low. Regarding positive examples, we explicitly instructed annotators and validators to identify similar or related queries and to cross-annotate documents accordingly. As a result, some queries share the same relevant documents.
>
> In the Appendix F.1 of the revised version, we have added a human evaluation to estimate the amount of false positive and negative documents. As shown in the Table below, the audit revealed zero false positives and a false negative rate of only 2.5% for Knowledge tasks. Similarly, for Theorem tasks, the combined error rate was minimal at approximately 5.8%, comprising 3.3% false negatives and 2.5% false positives. These results suggest that label noise is insignificant, thereby supporting the reliability of the benchmark.
>
> Table: Human evaluation of document relevance annotations for the top-retrieved documents produced by the best-performing multimodal model Ops-MM-Embedding. A false negative refers to a relevant document incorrectly labeled as irrelevant in ours, while a false positive refers to an irrelevant document incorrectly labeled as relevant.
>
> |               |                   | **False Negatives** |                    | **False Positives** |                    |
> |---------------|-------------------|---------------------|--------------------|---------------------|--------------------|
> | **Dataset**   | **Documents Checked** | **Count**           | **Ratio**          | **Count**           | **Ratio**          |
> | Theorem       | 120               | 4                   | 3.3%               | 3                   | 2.5%               |
> | Knowledge     | 120               | 3                   | 2.5%               | 0                   | 0.0%               |
>
>
>
> > 2. Smaller dataset compared to sme of the MTEB/MMEB datasets.
>
> Thanks for the comment. The MTEB/MMEB datasets are focusing on assembling a great amount of diverse tasks in one framework, which is not our primary goal. We aim to providing more challenging multimodal retrieval tasks in expert domains with human annotations. More importantly, we curate novel task formulations, queries, relevant documents involving a significant amount of human annotations which is missing in these datasets.
>
> As shown in Table 1, MRMR contains 1,435 expert-annotated queries across 23 domains. While this may appear small by NLP standards, in the information retrieval (IR) community, benchmarks with 30–50 queries are often considered valid and effective, since each query requires meticulous relevance assessment across many candidate documents [1][2]. This is the primary reason we curate approximately 100 queries for each subtask. Moreover, recent text and multimodal retrieval benchmarks typically contain a comparable number of queries. We summarize them below for reference.
>
> | Benchmark      | Modality                              | #Queries | Task / Focus                                                                 |
> | -------------- | -------------------------------------- | -------- | ----------------------------------------------------------------------------- |
> | FollowIR [1]   | Text                                   | 349      | Re-ranking documents according to user-provided instructions                  |
> | BRIGHT [3]     | Text                                   | 1,398    | Reasoning-intensive text retrieval (math, code, social science, etc.)        |
> | CIRR [4]       | Multimodal                             | 4,148    | Retrieve real-life images that match a text-guided modification of a reference image |
> | ViDoRe [5]     | Multimodal                             | 3,810    | Retrieve multimodal document pages containing text, figures, and tables   |
> | MMDocIR [6]    | Multimodal                             | 1,658    | Retrieve multimodal document pages containing text, figures, and tables           |
> | MR2-Bench [7]  | Multimodal                             | 1,309    | Reasoning-oriented multimodal retrieval                                      |
> | **MRMR (Ours)**| **Multimodal**| **1,502** | **Reasoning-intensive retrieval across Knowledge, Theorem, and Contradiction tasks** |

---

> ### Author Response · Authors · 2025-11-20
> **Weaknesses (2)**
>
> Importantly, MRMR is costly because, like BRIGHT [3] and MR2‑Bench [7], it requires domain experts (e.g., clinicians, engineers) to judge not just topical relevance but whether a document genuinely supports the reasoning needed to answer the query. Each query involves constructing or validating complex multimodal scenarios, sourcing relevant documents from the web, and distinguishing true evidence from hard negatives. This high per‑query cost limits scale but ensures high-quality, reasoning-intensive labels.
>
> [1] Weller, O., Chang, B., MacAvaney, S., Lo, K., Cohan, A., Van Durme, B., and Soldaini, L. 2024. FollowIR: Evaluating and Teaching Information Retrieval Models to Follow Instructions. Proceedings of the 2025 Conference of the Nations of the Americas Chapter of the Association for Computational Linguistics: Human Language Technologies (Volume 1: Long Papers).
>
> [2] Webber, W., Moffat, A., and Zobel, J. 2008. Statistical power in retrieval experimentation. In Proceedings of the 17th ACM Conference on Information and Knowledge Management (CIKM 2008).
>
> [3] Su, H., Yen, H., Xia, M., Shi, W., Muennighoff, N., Wang, H., Haisu, L., Shi, Q., Siegel, Z. S., Tang, M., Sun, R., Yoon, J., Arik, S. O., Chen, D., and Yu, T. 2025. BRIGHT: A Realistic and Challenging Benchmark for Reasoning-Intensive Retrieval. In Proceedings of the International Conference on Learning Representations (ICLR 2025).
>
> [4] Liu, Z., Rodriguez-Opazo, C., Teney, D., and Gould, S. 2021. Image retrieval on real-life images with pre-trained vision-and-language models. In Proceedings of the IEEE/CVF International Conference on Computer Vision (ICCV 2021).
>
> [5] Macé, Q., Loison, A., and Faysse, M. 2025. ViDoRe Benchmark V2: Raising the Bar for Visual Retrieval. arXiv preprint arXiv:2505.17166.
>
> [6] Dong, K., Chang, Y., Goh, X. D., Li, D., Tang, R., and Liu, Y. 2025. MMDocIR: Benchmarking Multimodal Retrieval for Long Documents. In Proceedings of the 2025 Conference on Empirical Methods in Natural Language Processing (EMNLP 2025).
>
> [7] Anonymous. 2025. MR²-Bench: Going Beyond Matching to Reasoning in Multimodal Retrieval. Submitted to International Conference on Learning Representations (ICLR 2025).
>
> > 3. Evaluation only benchmark. It would have been significantly more useful had a training dataset been provided.
>
> Thank you for the constructive feedback. In our annotation process, we observed that only about 60% of the documents labeled as relevant by GPT-5 were confirmed as relevant by human experts. In addition, GPT-Search failed to retrieve relevant webpages for nearly 50% of the queries, requiring human experts to manually create relevant documents. As a result, the current pipeline still depends heavily on human involvement, which limits its scalability for constructing large-scale training datasets. We plan to further refine the pipeline and explore the training methods  in future work.
>
> > 4. Query expansion results for stronger retriever models missing. It is important to understand if query exapansion with multimodal/text retriver models still falls short on this benchmark.
>
> Thanks for the insightful suggestion, We have added the query expansion results for the strong retriever model Ops-MM-Embedding in the Appendix F.3 of the revised version. With the comparison between weak and strong retrievers, we observe following new findings:
>
> - With query expansion by a weak LLM (Qwen2-VL-2B), the weaker retriever receives a performance boost but the performance of the stronger retriever degrades. It shows that the query expansion quality is of importance.
>
> -  With query expansion by a strong LLM (Qwen2.5-VL-72B), the expansion technique is effective for improving both strong and weak retriever models. However, they are still far from perfect on this benchmark. For example, the best retriever with strong query expansion only achieves 55.9 for medical queries and 31.8 for math queries.
>
> > 5. Weak qualitative analysis. A category wise qualitative analysis would have been much more helpful.
>
> Thanks for the suggestion. We have added more qualitative analysis details in Appendix F of the new version. Especially, in Appendix F.2, we have added error case studies for more different categories in addition to Biology and Traffic Case examples. Appendix F.1, we conduct human evaluation for false positive and false negative relevance annotations for documents.  In Appendix F.3, we add test-time scaling experiments for the strong retriever model Ops-MM-Embedding in addition to the weaker retriever model GME-Qwen2-VL.

---

### Official Review · Reviewer_u6P3 · 2025-11-05

**Soundness:** 3
**Presentation:** 3
**Contribution:** 3
**Rating:** 6
**Confidence:** 4

**Summary:**

The authors introduce a dataset focusing on 'Multidisciplinary and Reasoning-intensive Multimodal Retrieval' MRMR with many key contributions with respect to existing multi-modal retrieval dataset. From the perspective of somebody likely to utilize this dataset, there are two practical high-level contributions: (1) MRMR provides validated retrieval results for a subset of the MMMU-Pro question source based on high-quality datasources (PIN-14M + web pages for {Knowledge: Art, Medicine, Science, Humanities} and Bright + web pages for {Theorem: Math, Physics, Engineering, Humanities}; and (2) MRMR develops an interesting reasoning-intensive "contradiction" problem and dataset that is applicable to agentic LLM settings (and currently has low performance with existing systems). In contrasting this work with existing MMIR benchmarks, some key contributions of MRMR (also pointed out by the authors) include: (1) multi-disciplinary expert-level questions (most benchmarks focus on the general knowledge of Wikipedia), (2) reasoning-focus with more complex IR instructions (most benchmarks focus on semantic matching or text IR/QA), and (3) image-text interleaving in queries and data (most MM benchmarks are VQA-focused with a single image).

For each question type {Knowledge, Theorem, Contradiction}, the authors first describe their method for selecting questions, collecting/validating positive and hard negative results, and adding a background set of reasonable negative examples to construct the corpus. Based on the MRMR dataset, the authors conduct a large scale ablation study over four multi-modal retrieval setups (Text models with image captions, Text and image two-stream models with vector function, Multimodal model with merged images, and Multimodal models with document as image) with 14 state-of-the-art candidate models, demonstrating: (1) text-retrieval based approaches presently have the best performance for these knowledge/reasoning-required tasks, (2) there is ample room for improvement across the board, and (3) there is large variance in performance across different tasks. This study was able to power a reasonable failure-space analysis to point toward research directions that may result in tangible improvements.

**Strengths:**

Some strengths of this work include:
- As primarily a resources paper, the authors clearly defined what type of questions they targeting, clearly motivated why this would result in a useful benchmark, and demonstrated that a large-scale ablation with the benchmark surfaces interesting (albeit not entirely surprising) conclusions.
- The authors clearly contrast MRMR with related datasets (Table 1) and clearly describe the collection procedure in the various settings (Table 2 and Sections 3.2-3.4) -- which are generally sensible and required a significant amount of work to collect with high-quality.
- MMMU-Pro is a good dataset to derive challenging IR problems with.
- The 'contradiction' dataset is innovative and I expect will inspire additional work in other challenges IR cases that require specific instructions and will lead to more interesting agentic LLM responses.
- A good benchmark paper introduces a non-trivial dataset, makes it easy for others to build on, and provides non-trivial baseline performance results -- which MRMR does.

**Weaknesses:**

Some weaknesses of this work include:
- The 'contradiction' family of questions is the most interesting from an innovation perspective, but underdeveloped in my opinion. To begin with, even if I am being a bit pedantic, these technically aren't all contradictions (e.g., Figure 1 vehicle design is 'non-compliance'). Also, in looking through the specific cases, 'negation' is a bit contrived (albeit shown to perform poorly in the empirical results) and I can think of other cases and possibly a general hierarchy of "matching a description that isn't there". Basically, it is somewhat preliminary and likely deserves its own study.
- The dataset is high-quality, but relatively small as compared to other related datasets in Table 1. Minimally, I would recommend discussing why a smaller number of high-quality examples is more valuable with respect to detecting improvements.
- Based on the process used, it wasn't clear if there would be any license issues with respect to using this in commercial settings (even if only for academic purposes). I am assuming this isn't an issue, but it would likely affect its impact.
- The analysis in Section 5 is underdeveloped.

**Questions:**

- It would be useful for you to clarify any licensing issues for academic work in commercial settings.
- In terms of the ablation study, it would also be interesting in the discussion to point out any alignment or contradictions for similar systems on related datasets (e.g., wikiHow).

---

> ### Author Response · Authors · 2025-11-20
> **Weaknesses (1)**
>
> We sincerely thank the reviewer for their positive feedback and for recognizing our work as a solid, nontrivial, and original contribution. Below, we address the concerns they raised.
>
> > 1. The 'contradiction' family of questions is the most interesting from an innovation perspective, but underdeveloped in my opinion. To begin with, even if I am being a bit pedantic, these technically aren't all contradictions (e.g., Figure 1 vehicle design is 'non-compliance'). Also, in looking through the specific cases, 'negation' is a bit contrived (albeit shown to perform poorly in the empirical results) and I can think of other cases and possibly a general hierarchy of "matching a description that isn't there". Basically, it is somewhat preliminary and likely deserves its own study.
>
> Thank you for the thoughtful comments. We agree that Contradiction Retrieval can be further expanded and explored. Our goal in this work is to define **a setting where the gold document is opposite or conflicting with the query**. Prior retrieval benchmarks mostly focus on queries whose gold documents are positively aligned, leaving the "oppositional" setting underexplored.
>
> We acknowledge that contradiction is a broad concept with many possible interpretations. In our current formulation, we specifically narrow it to cases where the query and document express conflicting or opposite statements. This design was inspired by prior work on negation, though our reframing into a more general "conflict" setting is admittedly somewhat contrived. In contrast, the Traffic Case and Vehicle Design tasks adopt a more realistic paradigm, i.e., retrieving the violated rule or principle, offering a more naturally grounded form of contradiction.
>
> In future work, we could consider a more dedicated and detailed study of contradiction-based retrieval, including richer typologies of conflicting relations and more systematic query–document constructions.
>
> > 2. The dataset is high-quality, but relatively small as compared to other related datasets in Table 1. Minimally, I would recommend discussing why a smaller number of high-quality examples is more valuable with respect to detecting improvements.
>
>
> Thanks for the comment. As shown in the table below, MRMR contains 1,435 expert-annotated queries across 23 domains. While this may appear small by NLP standards, in the information retrieval (IR) community, benchmarks with 30–50 queries are often considered valid and effective, since each query requires meticulous relevance assessment across many candidate documents [1][2]. This is the primary reason we curate approximately 100 queries for each subtask. Moreover, recent text and multimodal retrieval benchmarks typically contain a comparable number of queries. We summarize them below for reference.
>
> | Benchmark      | Modality                              | #Queries | Task / Focus                                                                 |
> | -------------- | -------------------------------------- | -------- | ----------------------------------------------------------------------------- |
> | FollowIR [1]   | Text                                   | 349      | Re-ranking documents according to user-provided instructions                  |
> | BRIGHT [3]     | Text                                   | 1,398    | Reasoning-intensive text retrieval (math, code, social science, etc.)        |
> | CIRR [4]       | Multimodal                             | 4,148    | Retrieve real-life images that match a text-guided modification of a reference image |
> | ViDoRe [5]     | Multimodal                             | 3,810    | Retrieve multimodal document pages containing text, figures, and tables   |
> | MMDocIR [6]    | Multimodal                             | 1,658    | Retrieve multimodal document pages containing text, figures, and tables           |
> | MR2-Bench [7]  | Multimodal                             | 1,309    | Reasoning-oriented multimodal retrieval                                      |
> | **MRMR (Ours)**| **Multimodal**| **1,435** | **Reasoning-intensive retrieval across Knowledge, Theorem, and Contradiction tasks** |
>
> Importantly, MRMR is costly because, like BRIGHT [3] and MR2‑Bench [7], it requires domain experts (e.g., clinicians, engineers) to judge not just topical relevance but whether a document genuinely supports the reasoning needed to answer the query. Each query involves constructing or validating complex multimodal scenarios, sourcing relevant documents from the web, and distinguishing true evidence from hard negatives. This high per‑query cost limits scale but ensures high-quality, reasoning-intensive labels.

---

> ### Author Response · Authors · 2025-11-20
> **Weaknesses (2)**
>
> [1] Weller, O., Chang, B., MacAvaney, S., Lo, K., Cohan, A., Van Durme, B., and Soldaini, L. 2024. FollowIR: Evaluating and Teaching Information Retrieval Models to Follow Instructions. Proceedings of the 2025 Conference of the Nations of the Americas Chapter of the Association for Computational Linguistics: Human Language Technologies (Volume 1: Long Papers).
>
> [2] Webber, W., Moffat, A., and Zobel, J. 2008. Statistical power in retrieval experimentation. In Proceedings of the 17th ACM Conference on Information and Knowledge Management (CIKM 2008).
>
> [3] Su, H., Yen, H., Xia, M., Shi, W., Muennighoff, N., Wang, H., Haisu, L., Shi, Q., Siegel, Z. S., Tang, M., Sun, R., Yoon, J., Arik, S. O., Chen, D., and Yu, T. 2025. BRIGHT: A Realistic and Challenging Benchmark for Reasoning-Intensive Retrieval. In Proceedings of the International Conference on Learning Representations (ICLR 2025).
>
> [4] Liu, Z., Rodriguez-Opazo, C., Teney, D., and Gould, S. 2021. Image retrieval on real-life images with pre-trained vision-and-language models. In Proceedings of the IEEE/CVF International Conference on Computer Vision (ICCV 2021).
>
> [5] Macé, Q., Loison, A., and Faysse, M. 2025. ViDoRe Benchmark V2: Raising the Bar for Visual Retrieval. arXiv preprint arXiv:2505.17166.
>
> [6] Dong, K., Chang, Y., Goh, X. D., Li, D., Tang, R., and Liu, Y. 2025. MMDocIR: Benchmarking Multimodal Retrieval for Long Documents. In Proceedings of the 2025 Conference on Empirical Methods in Natural Language Processing (EMNLP 2025).
>
> [7] Anonymous. 2025. MR²-Bench: Going Beyond Matching to Reasoning in Multimodal Retrieval. Submitted to International Conference on Learning Representations (ICLR 2025).

---

> ### Author Response · Authors · 2025-11-20
> **Weaknesses (3)**
>
> > 3. Based on the process used, it wasn't clear if there would be any license issues with respect to using this in commercial settings (even if only for academic purposes). I am assuming this isn't an issue, but it would likely affect its impact.
>
> Thanks for raising the licensing concern. Following the practice of MMMU, our annotators and validators are instructed to avoid using materials from websites that prohibit copying or redistribution when reviewing MRMR documents. Consequently, most MRMR documents are derived from sources that are free of copyright restrictions, such as Wikipedia pages, government reports (e.g., NIH and Singapore Police Force), and PubMed Central (PMC). The datasets we build upon also carry permissive public licenses, including MMMU (Apache-2.0), PIN-14M (CC-BY-4.0), COCO (CC-BY-4.0), and BRIGHT (CC-BY-4.0). We have added these clarifications to the Ethics Statement section.
>
> > 4. The analysis in Section 5 is underdeveloped.
>
> Thanks for pointing out. We have added more analysis details in Appendix F of the new version. In Appendix F.1, we conduct human evaluation for false positive and false negative relevance annotations for documents. In Appendix F.2, we add more case studies for different domains. In Appendix F.3, we add test-time scaling experiments for the strong retriever model Ops-MM-Embedding in addition to the weaker retriever model GME-Qwen2-VL.

---

> ### Author Response · Authors · 2025-11-20
> **Questions**
>
> > 1. It would be useful for you to clarify any licensing issues for academic work in commercial settings.
>
> As discussed in the Weakness, we have added clarifications of licensing issues in the CODE OF ETHICS AND ETHICS STATEMENT section in the new version.
>
> > 2. In terms of the ablation study, it would also be interesting in the discussion to point out any alignment or contradictions for similar systems on related datasets (e.g., wikiHow).
>
> We appreciate the reviewer’s suggestion to relate MRMR more concretely to prior retrieval work, especially wikiHow [1]. In the revised manuscript, we have added the discussion and a comparison table summarizing the performance of overlapping systems on both benchmarks. Specifically, several models are evaluated on both wikiHow and MRMR, including VISTA, E5‑V, MM‑Embed, GME‑Qwen2‑VL, and Jina‑CLIP‑v2.
>
>
> | Model              | wikiHow (overall) | MRMR Knowledge (avg) | MRMR Theorem (avg) | MRMR Contradiction (avg) | MRMR Overall (avg) |
> |--------------------|------------------------|-----------------------|---------------------|---------------------------|---------------------|
> | VISTA              | 35.2                   | 24.7                  |   20.5           | 16.5                      | 20.9              |
> | E5‑V               | 50.5                   | 16.1                  |  3.3              |  5.8                      |  8.6               |
> | MM‑Embed           | 56.4                   | 61.2                  | 31.7            | 21.9                      | 39.8         |
> | GME‑Qwen2‑VL       | 54.1                   | 46.7                  |  35.0               | 23.6                      | 36.2               |
> | Jina‑CLIP‑v2       | 47.2                   | 19.0                  |  10.3              | 12.2                      | 14.0            |
>
>
> From this cross‑benchmark view we observe:
>
> **Alignment in model performance ranking.** The relative ordering of systems is broadly consistent: models that are strong on wikiHow (e.g., MM‑Embed, GME‑Qwen2‑VL) are also among the best multimodal retrievers on MRMR, whereas weaker models (e.g., Jina‑CLIP‑v2) remain weaker on both datasets.
>
> **Performance drop on MRMR reasoning tasks.** At the same time, absolute scores are much lower on MRMR—especially for Theorem and Contradiction. For instance, E5‑V drops from 50.5 nDCG@10 on wikiHow to 3.3 on MRMR Theorem and 5.8 on MRMR Contradiction. Even MM‑Embed, which remains relatively strong, falls from 56.4 on wikiHow to 31.7 (Theorem) and 21.9 (Contradiction).
>
> These results indicate that wikiHow and MRMR are consistent but complementary: wikiHow primarily probes interleaved modeling and semantic matching in general‑domain how‑to tutorials, whereas MRMR exposes substantial additional difficulty from expert‑level, reasoning‑intensive retrieval in medicine, science, engineering, and law‑like rule corpora.
>
> [1] Xin Zhang, Ziqi Dai, Yongqi Li, Yanzhao Zhang, Dingkun Long, Pengjun Xie, Meishan Zhang, Jun Yu, Wenjie Li, and Min Zhang. 2025. Towards Text-Image Interleaved Retrieval. In Proceedings of the 63rd Annual Meeting of the Association for Computational Linguistics.

---

### Author Response · Authors · 2025-11-30
**Official Comment by Authors**

**To all reviewers:**

We sincerely thank all reviewers for their time and thoughtful assessment of our manuscript. We have carefully addressed every concern through point-by-point responses, and incorporated the corresponding clarifications into the revised version.

We are encouraged by the positive scores and feedback from all reviewers—particularly regarding the novelty and significance of our benchmark (**Reviewers u6P3, 8xXt, 8qw3, and 9XX3**), the broad and meaningful domain coverage and human annotations (**Reviewers u6P3, 8qw3, and 9XX3**), the comprehensive evaluation (**Reviewers u6P3 and 9XX3**), and the clarity of our writing (**Reviewer 8xXt**).

The main concerns by reviewers and our responses are as follows:

- **Limited dataset size**: Our target scale follows recent text and multimodal Information Retrieval (IR) benchmarks (e.g., BRIGHT, MMDocIR) that contain ~1k queries. In IR communities, 30–50 well-annotated queries are often considered sufficient due to the heavy relevance-assessment workload per query.

- **Preliminary scope of “Contradiction Retrieval”**: As a first multimodal formulation of contradiction-based retrieval, we narrow the task to clear conflicting or opposite statements. It could require more comprehensive work, deserving its own study.

- **Data license issues**: Following MMMU, annotators are instructed to avoid materials from websites that prohibit copying or redistribution. Most MRMR documents therefore originate from copyright-permissive sources, and we have added a clarification in the paper.

- **False negative and false positive documents**: Annotators actively check for similar queries to reduce false negatives. We added a human evaluation study showing low error rates for both false positives and false negatives.

- **Evaluation-only data without training data**: The low success rate of GPT-Search and limited agreement between GPT-5 and human experts currently necessitate substantial human involvement. This limits scalability, so we currently restrict MRMR to an evaluation benchmark.

In addition, in the revised version, we have added the query expansion experiments for the stronger retriever and corresponding analysis. To improve qualitative analysis, we have added more error cases across domains. Moreover, we identified the unqualified queries in the Theorem Tasks. Thus, we have filtered out these subtasks and updated the evaluation results.

For more details, please refer to our responses to each reviewer. We have strived to address each of your concerns.

Sincerely yours,

Authors

---

### Meta-Review · Area_Chair_Pehh · 2025-12-22

**Summary:**

As the authors summarized, the main concerns from the reviewers are summarized as follows:

Dataset Size: The given dataset size aligns with recent benchmarks in text and multimodal Information Retrieval (IR), such as BRIGHT and MMDocIR, which contain around 1,000 queries. In the IR community, 30–50 well-annotated queries are often considered sufficient, given the significant workload involved in relevance assessment for each query.

The Scope of “Contradiction Retrieval”: As the first multimodal approach to contradiction-based retrieval, this paper focuses on conflicting or opposite statements.

Data License Issues: In line with MMMU guidelines, annotators are instructed to avoid using content from websites with restrictive copyright terms.

False Negative and False Positive Documents: Annotators actively cross-check similar queries to minimize false negatives.

Evaluation-Only Data Without Training Data: Due to the low success rate of GPT-Search and the limited agreement between GPT-5 and human experts, the process currently requires significant human involvement.

**Reviewer Concerns:**

The authors well addressed the concerns from different reviewers.

**Reviewer Scores:**

All the reviewers held positive comments about this paper in the original evaluations.

---

### Decision · Program_Chairs · 2026-01-26

Accept (Poster)